# Path Channels and Plan Extension Kernels: a Mechanistic Description of Planning in a Sokoban RNN

**Mohammad Taufeeque, Aaron David Tucker, Adam Gleave & Adrià Garriga-Alonso**
FAR.AI
Berkeley, California, United States of America
`{taufeeque,adria}@far.ai`

## Abstract

We partially reverse-engineer a convolutional recurrent neural network (RNN) trained with model-free reinforcement learning to play the box-pushing game Sokoban. We find that the RNN stores future moves (plans) as activations in particular channels of the hidden state, which we call *path channels*. A high activation in a particular location means that, when a box is in that location, it will get pushed in the channel's assigned direction. We examine the convolutional kernels between path channels and find that they encode the change in position resulting from each possible action, thus representing part of a learned *transition model*. The RNN constructs plans by starting at the boxes and goals. These kernels, *extend* activations in path channels forwards from boxes and backwards from the goal. Negative values are placed in channels at obstacles. This causes the extension kernels to propagate the negative value in reverse, thus pruning the last few steps and letting an alternative plan emerge; a form of backtracking. Our work shows that, a precise understanding of the plan representation allows us to directly understand the bidirectional planning-like algorithm learned by model-free training in more familiar terms.

## 1 Introduction

Modern AI systems can accomplish a wide variety of complicated tasks, but neural networks trained using deep learning are often difficult to understand. This may be concerning in the context of agentic behavior, where the AI takes a complicated sequence of actions to accomplish a goal. In such cases, agents may accomplish potentially challenging tasks in ways that are difficult to anticipate, making the consequences of their actions hard to foresee. Can the behavior of machine-learned agents be understood? This work answers in the affirmative for the case of an agent trained to play the puzzle game Sokoban using model-free reinforcement learning.

We study Sokoban due to consensus in the literature that a particular architecture exhibits planning behavior, forming an internal representation of its anticipated future states and actions which can be extracted using linear probes. Sokoban is a grid-based puzzle game with walls ■, floors □, movable boxes ■, and target tiles ■ where the agent's goal ■ is to push all boxes onto target tiles. Since boxes can be pushed (not pulled), wrong moves can make the puzzle unsolvable, making Sokoban a challenging game that is PSPACE-complete (Culberson, 1997), requires long-term planning, and a popular planning benchmark (Peters et al., 2023; Racanière et al., 2017; Hamrick et al., 2021). Guez et al. (2019) introduced the DRC architecture family and showed that DRC$(3, 3)$ achieves state-of-the-art performance on Sokoban amongst model-free RL approaches and rival model-based agents like MuZero (Schrittwieser et al., 2019; Chung et al., 2024). They argue that the network exhibits *planning* behavior since it is data-efficient in training, generalizes to multiple boxes, and benefits from additional compute. Specifically, the solve rate of the DRC improves by 4.7% when the network is fed the first observation ten times during inference, giving the network extra thinking time.Bush et al. (2025) use logistic regression probes to find a causal representation of the plan in the DRC, and qualitatively suggest, based on these representations, that the algorithm performs bidirectional search. Taufeeque et al. (2024) find that the DRC often pauses for a few steps before

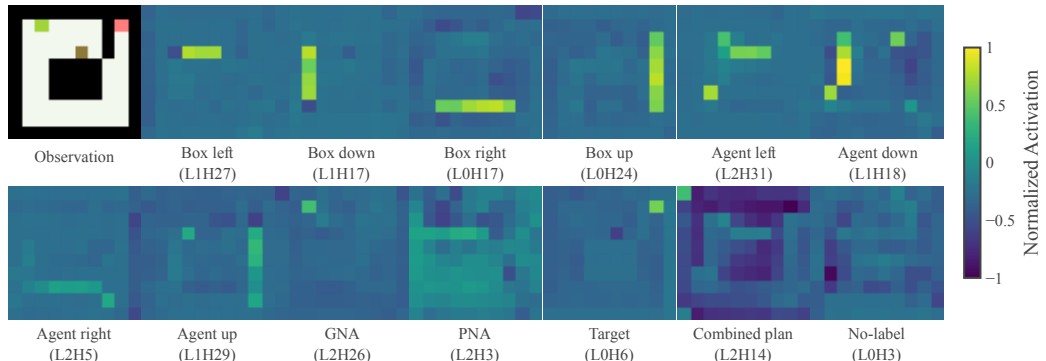

Figure 1: A level with channel activations for a single channel from every group in Table 1. Note the clear activations for the box move channels along the box ■'s path to the target ■.

acting and that the plan changes more quickly during those steps, indicating that the policy seeks test-time compute when needed.

Our contributions are to substantially reverse-engineer the representation that the DRC agent uses to represent its future states and actions, as well as the mechanisms which construct its plan. We find that many channels of the DRC$(3, 3)$ network directly represent the propensity to move in a given direction in the short or long term, calling them *path channels*. This removes the need for linear probes used in prior work (Bush et al., 2025; Taufeeque et al., 2024), reducing the analysis to simply reading channel representations. We then analyze convolutional kernels, as well as activations' evolution over time, to mechanistically understand the agent's planning algorithm. The paper analyses one single seed of DRC$(3, 3)$ with Section N showing that the results replicate on four other independently trained networks as well.

## 2 THE PLAN REPRESENTATION

In contrast to the linear probes discovered by earlier works, we radically simplify the representation. Hidden state channel activations in the DRC$(3, 3)$ architecture from Guez et al. (2019) directly represent the propensity of the agent or box to move in particular direction.

### 2.1 NETWORK ARCHITECTURE

We analyze the open-source DRC$(3, 3)$ network trained by Taufeeque et al. (2024) to solve Sokoban, who closely followed the training setup of Guez et al. (2019). The network consists of a convolutional encoder, a stack of 3 ConvLSTM layers, and an MLP head for the policy and RL value function prediction. Each ConvLSTM block perform 3 ticks of recurrent computation per in-game timestep. The encoder block $E$ consists of two $4 \times 4$ convolutional layers without nonlinearity, which process the $H \times W \times 3$ RGB observation $x_t$ into an $H \times W \times C$ output $e_t$ with height $H$, width $W$, and $C$ channels, at environment step $t$.

**ConvLSTM Layers.** Figure 2b visualizes the computation of the ConvLSTM layer. Each of the ConvLSTM layers at depth $d$ and tick $n$ in the DRC maintains hidden states $h_d^n, c_d^n$ with dimensions $H \times W \times C$ and takes as input the encoder output $e_t$, the previous layer's hidden state $h_{d-1}^n, c_d^{n-1}$, and its own hidden state $h_d^{n-1}$ from the previous step. The ConvLSTM layer computes four parallel gates $i, j, f, o$ using convolutional operations with $3 \times 3$ kernels that are combined to update the hidden state. For the first ConvLSTM layer ($d = 0$), the architecture uses the top-down skip connection from the last ConvLSTM layer ($d = 2$). This gives the network $3 \cdot 3 = 9$ layers of sequential computation to determine the next action at each step. The final layer's hidden state at the last tick is processed through an MLP head to predict the next action and value function.

**DRC$(3, 3)$ architecture.** The DRC$(3, 3)$ architecture applies the ConvLSTM modules as shown in Figure 2, encoding the observations $x_t$ then applying the ConvLSTM layers parameterized by $\theta_1, \theta_2$,

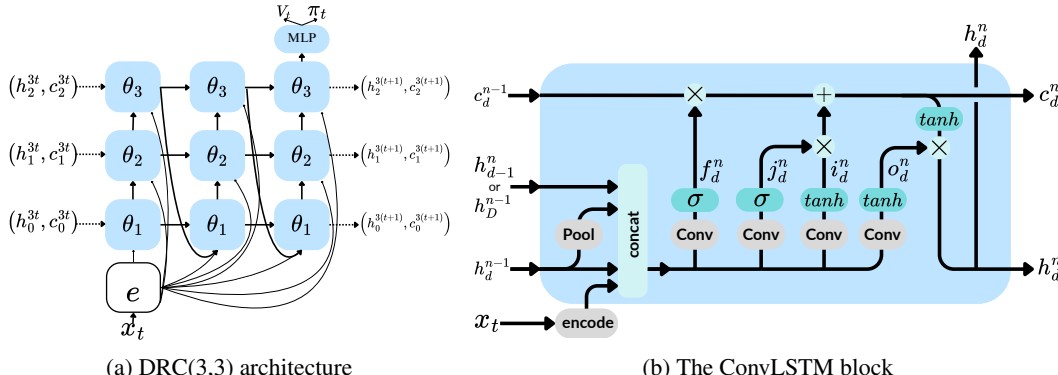

(a) DRC(3,3) architecture          (b) The ConvLSTM block

Figure 2: **Left**: The DRC$(3,3)$ architecture. There are three layers of ConvLSTM modules with all the layers repeatedly applied three times before predicting the next action. **Right**: The ConvLSTM block in the DRC. Note the use of convolutions instead of linear layers, and the last layer of the previous tick ($h_D^{n-1}$) as input to the first layer. "Pool" refers to a weighted combination of mean- and max-pooling.

and $\theta_3$ in sequence for three layers, then applying the same modules to the previous hidden states (i.e. the hidden states from the previous layer and the previous ticks Figure 2b) three times for three "ticks". The entire network is run three times to choose the action in response to each observation.

**Notation.** Each DRC tick ($n = 0, 1, 2$) involves three ConvLSTM layers ($d = 0, 1, 2$), each providing six 32-channel tensors ($h_d, c_d, i_d, j_d, f_d, o_d$). Channel $c$ of tensor $v_d$ is denoted L$d$V$c$.

**Additional details.** Additional architecture, training, and dataset details provided in Sections B and C. Unless mentioned otherwise, the dataset of levels for all results is the medium-difficulty validation set from Boxoban (Guez et al., 2018).

## 2.2 PATH CHANNELS

With the architecture and notation defined, we now describe how the DRC$(3,3)$ agent plan is representing in *path channels* of the hidden states $h_0, h_1$, and $h_3$.

At each layer, the DRC has $C$ channels, each of which is a $H \times W$ array. The DRC repeats the same computations convolutionally over each square. This results in a subset of hidden state channels $h_d$ representing the plan where each channel corresponds to a movement direction. If the agent or a box is at a position where the channel is activated, this causes the DRC to choose that channel's direction as the action. Section 2.2 illustrates box path channels, where the box ■ moves down twice then right twice to reach the target ■. Since the path channels are hidden convolutional states, each hidden state is repeated for the $H \times W$ game grid, with the 6 squares extending behind the game representing channels $C$. In the figure, the blue down channels are activated at the two locations where the box ■ moves down, and the purple right channels are activated at the two locationns where the box moves right. This figure has only one down channel and one right channel, while the real DRC$(3,3)$ network has several.

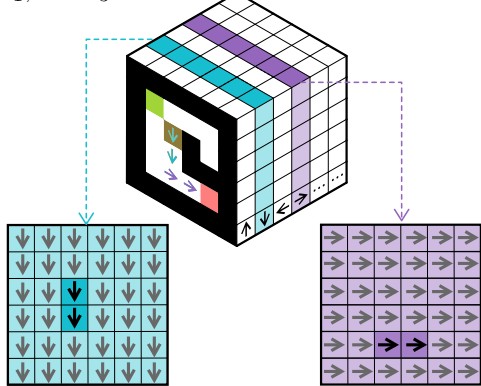

Figure 3: An idealized path channels diagram.

Manual inspection of every channel across all layers revealed that most channels are interpretable (Table 1). Detailed labels are in Tables 8 and 9 of the Appendix. We group the channels:

Table 1: Channel groups, their definitions and counts for each direction (up, down, left, right).

| Group | Definitions | Channels |
|---|---|---|
| Box-movement | Path of box (short- and long-term) | 20 (3, 6, 5, 6) |
| Agent-movement | Path of agent (short- and long-term) | 10 (3, 2, 1, 4) |
| Grid Next Action (GNA) | Immediate next action, represented at agent square | 4 (1, 1, 1, 1) |
| Pooled Next Action (PNA) | Pools GNA to represent next action in all squares | 4 (1, 1, 1, 1) |
| Entity | Target, agent, or box locations | 8 |
| Combined path | Aggregate 2+ directions from movement channels | 29 |
| No label | Difficult to interpret channels | 21 |

*1) Box-movement* and *2) Agent-movement* channels that, for each cardinal direction, activate highly on a square in the grid if the box or agent moves in that direction from that square at a future timestep. The probes from Taufeeque et al. (2024); Bush et al. (2025) aggregated information from these channels.

*3) Combined path* channels that aggregate directions from the box- and agent-movement channels.

*4) GNA channels* that extract the next action from the previous groups of channels.

*5) PNA channels* that pool the GNA channels to be picked up by the MLP to predict the next action.

*6) The entity channels* that predominantly represent target ▇ locations, with some also representing box ▇ and agent ▇ locations.

*7) 'no label' channels* for which we could not discern a pattern.

We define the *path channels* as the set comprising the box-movement, agent-movement, and combined path categories, as they collectively maintain the complete plan of action for the agent. The remaining groups (GNA, PNA, entity, and no-label) are termed *non-path channels*, storing primarily short-term information, with some state for move selection heuristics.

## 2.3 QUANTITATIVE EVIDENCE

**Ablating the state.** First, we tested our classification of channels as being path channels or non-path channels by performing a single-step cache ablation. Specifically, we set the relevant hidden state channel's previous value to zero, and recompute its activation based on the previous observation only. This removes long-term information from the ablated channel by zeroing it out, while still preserving information from short-term computations based on the encoder and non-ablated channels. This keeps the ablated channel's activations more in line with what the network encounters during training, while removing the ablated channel's ability to store plan information across multiple timesteps. Intervening on the 59 path channels caused a substantial $57.6\% \pm 2.8\%$ drop in the solve rate. By contrast, intervening on the 37 non-path channels resulted in a $10.5\% \pm 1.9\%$ performance decrease ((a significantly smaller, yet non-negligible, decrease). Controlling for channel count, intervening on a random subset of 37 path channels still led to a $41.3\% \pm 2.4\%$ drop in solve rate. This evidence strongly suggests that the computations essential for long-term planning on difficult levels are primarily carried out within the identified path channels.

**Area under the curve analysis.** Second, we find that each path channel is predictive of the box or agent's future movements. Figure 4 shows the AUC for using the box or agent path channels for predicting the box or agent movement for different numbers of timesteps in the future. Note that the path channels further decompose into short- and long-term channels (Table 8, Section K).

In Section N we use the AUC values as the basis for a simple automatic labeling method which corroborates our findings on 4 additional random training seeds.

**Long-term path channels.** The network utilizes the *long-term* channels to manage spatially overlapping plans for different boxes intended for different times. Figure 5 illustrates this: in cases where two boxes pass through the same square sequentially in different directions with the first box moving at $t = 0$, the long-term channel for the *second* move activates well in advance ($t \ll 0$), while

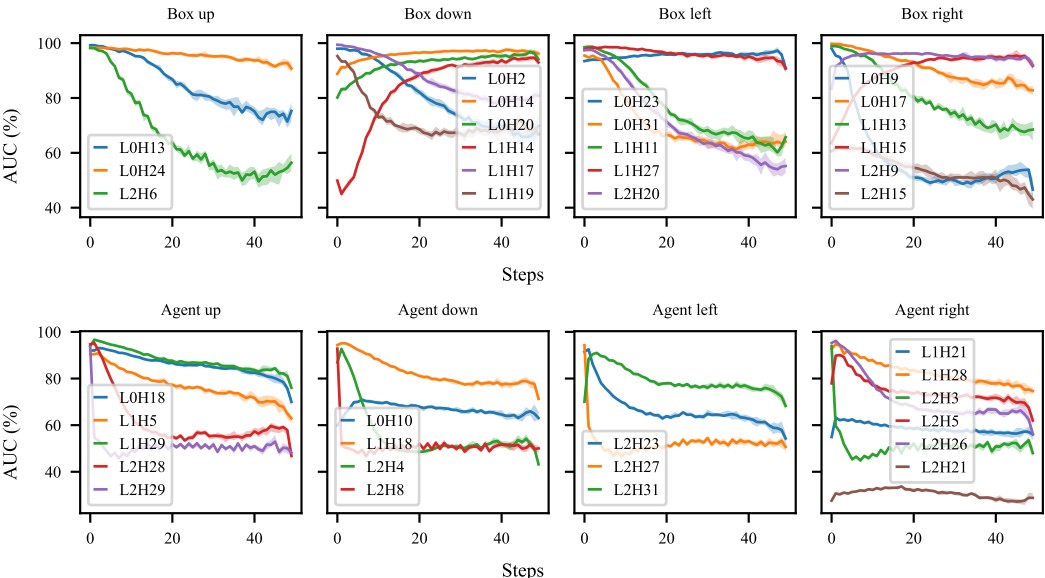

Figure 4: AUC scores for predicting box and agent movements from the path channels at different #s of timesteps out. Short-term channels have high AUC for up to 10 steps, while long-term channels show a high AUC for predicting actions beyond 10 steps until the end of the episode. The GNA/PNA path channels only exist for the agent, and have high AUC (∼100%) for only the next action.

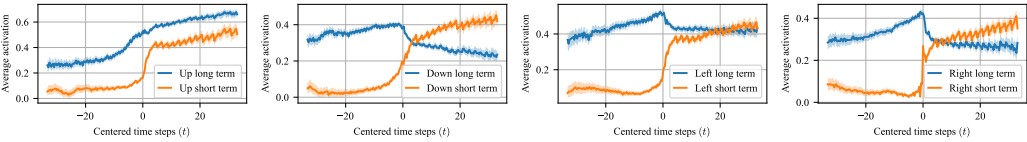

Figure 5: Activations of the long- and short-term channels for all directions when a different direction action takes place at $t = 0$. All directions except the up direction show the long-term channel activations decreasing after the other action takes place at $t = 0$. The mechanism of this transfer of activation from long to short-term is shown in Figure 15.

its corresponding short-term channel only becomes active after the first move is completed ($t = 0$). Figure 15 shows the mechanism of this transfer is primarily mediated through the $j$-gate.

**Causal intervention**   Finally, we tested the causal effect of modifying path channel activations.

We verify the channel labels by performing causal interventions on the channels. We modify the channel activations based on their labels to make the agent take a different action than the one originally predicted by the network. We collect a dataset of 10,000 transitions by running the network on the Boxoban levels (Guez et al., 2018), measuring the fraction of transitions where the intervention succeeds at causing the agent to take any alternate target action, following the approach of Taufeeque et al. (2024). Table 2 shows high intervention scores for every group except the agent-movement channels, with the PNA

Table 2: Causal intervention scores for different channel groups.

| Group | Score (%) |
|---|---|
| Pooled Next Action (PNA) | $99.7 \pm 0.2$ |
| Grid Next Action (GNA) | $98.9 \pm 0.4$ |
| Box- and agent-movement | $88.1 \pm 1.9$ |
| Box-movement | $86.3 \pm 2.1$ |
| Agent-movement | $53.2 \pm 2.1$ |
| Probe: box movement | $82.5 \pm 2.5$ |
| Probe: agent movement | $20.7 \pm 0.7$ |

interventions achieving state of the art causal intervention scores as compared to the probe-based approaches in Taufeeque et al. (2024) and Bush et al. (2025).

The lower score for agent-movement channels is because they are causally relevant only when the agent is not pushing a box, which we did not filter for. Section E further validates our channel labels.

**Conclusion.** We thus conclude that the network primarily represents its plan in the activations of the identified box-movement and agent-movement channels. These plans are then mapped to the next action through the group next action and pooled next action channels, explained in Section F.

# 3 THE PLANNING ALGORITHM

How does the plan get constructed? The plan is made by extending path segments forward from the boxes and agent, and backward from the targets, implementing bidirectional search as qualitatively observed by Bush et al. (2025). The fact that path channels directly represent paths in individual activations allows us to directly examine weight matrices in order to understand key components of the algorithm.

First, the network uses kernels which respond to visual features of the game to initialize path segments by activating path channels adjacent to key objects such as the agent ▇, box ▇, and target ▇. Second, plan extension kernels extend these path segments until encountering obstacles which are represented as negative path channel activations. The plan extension kernels can propagate these negative activations backward along path segments to prune paths which encounter problems. Finally, a winner-takes-all mechanism inhibits the activations of conflicting path segments in favor of path segments with stronger activations.

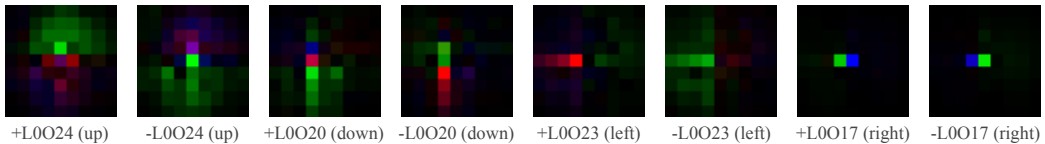

+L0O24 (up)   -L0O24 (up)   +L0O20 (down)   -L0O20 (down)   +L0O23 (left)   -L0O23 (left)   +L0O17 (right)   -L0O17 (right)

Figure 6: Visualizations of combined kernels that map from the RGB input to the $o$-gate of the up, down, left, and right box-movement channels of layer 0. The negative and positive RGB components are visualized separately. The kernels activate squares along (for agent ▇ and box ▇) and against (for target ▇) the channel's direction. These kernels when applied on the RGB input activate (initialize) few squares around the agent or box for forward plan chains and around the target for backward plan chains. The kernel for L0O17 (right) initializes plan chains only on the agent and box square.

## 3.1 INITIALIZING PATH CHANNEL ACTIVATIONS

Analysis of simplified encoder kernels mapping from the game observations to box movement path channels show structures that initialize path channel activations. These kernels detect relevant features (such as targets, boxes, or the agent's position) to add activations to initialize path segments, such as moving towards targets or away from boxes.

**Encoder Simplification.** We simplify the encoders for visualization purposes. Individually, the encoder weights have no privileged basis (Elhage et al., 2023). To interpret the weights of the encoder, we use the associativity of linear operations to combine the convolution operations of the encoder output $e_t$ at time $t$ to the output gate $o_d^n$ for layer $d$ tick $n$ into a single convolutional layer. For all layers $d \in \{1, 2, 3\}$ and tick $n \in \{1, 2, 3\}$ we define the combined kernel $A_{oe}^d$ and bias $b_{oe}^d$ as:

$$o_d^n = \tanh(W_{oe}^d * e_t + \text{other terms}) \qquad \text{LSTM Equation (9)}$$
$$e_t = W_{E_2} * (W_{E_1} * x_t + b_{E_1}) + b_{E_2} \qquad \text{LSTM Equation (3)}$$
$$W_{oe}^d * e_t = W_{oe}^d * (W_{E_2} * (W_{E_1} * x_t + b_{E_1}) + b_{E_2}) \qquad (1)$$
$$= A_{oe}^d * x_t + b_{oe}^d \qquad \text{up to edge effects (2)}$$

for $A_{oe}^d = W_{oe}^d W_{E_2} W_{E_1}$ and $b_{oe}^d = W_{oe}^d (W_{E_2} b_{E_1} + b_{E_2})$.

This results in $9 \times 9$ convolution kernels mapping observations to each gate (Figures 6 and 24).

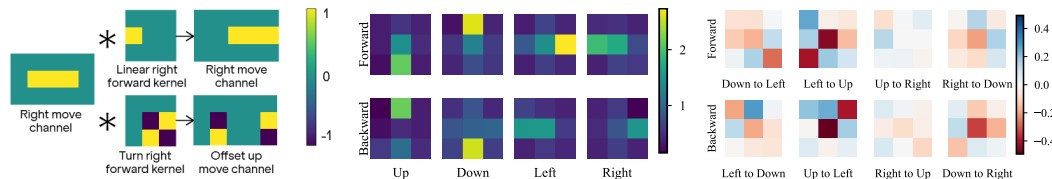

(a) Idealized path extension kernels. Convolutions extend moves in the relevant direction.

(b) Empirical linear plan extension kernels. Note reverse between forward and backward kernels.

(c) Empirical turn plan extension kernels. Compare to "turn right forward kernel" of subfigure (a).

Figure 7: Empirical plan extension kernels are formed by averaging over all kernels $W_{.h_2}$ mapping from the previous hidden state $h_d^{n-1}$ to each hidden state $\cdot \in \{i, j, f, o\}$ for each hidden state $\cdot$.

## 3.2 Extending Path Segments

**Plan Extension Kernels.** While the encoder kernels initialize path channel activations, turning these initial moves into path segments requires an extension mechanism operating on the recurrent hidden states. This is accomplished by specialized "plan-extension kernels" within the recurrent weight matrices ($W_{.h1}^d$ and $W_{.h_2}^d$ in Section B). *Linear plan extension kernels* (Figure 7b) propagate the plan linearly, extending it one square at a time along the channel direction label. Separate kernels exist to facilitate both forward chaining from boxes and backward chaining from targets. *Turn Plan Extension kernels* (Figure 7c) propagate activations from one channel to another channel representing a different direction. The linear kernels have larger weight magnitudes compared to the turn kernels, thus encoding agent's preference to turn only when unable to continue in the previous direction.

Figures 25 and 26, show some disaggregated linear and turn extension kernels. The idealized pattern definitely recurs in many kernels, and most have a pattern which is not the same, but is only translated or adds a square or two. In Section N we find plan extension kernels on four other training seeds.

**Weight steering.** While the agent is only trained on $10 \times 10$ boards, in Section L we use our knowledge of the path extension kernels to get the agent to solve a $40 \times 40$ level using weight steering. Since plans are represented in path channels, and constructed by the path extension kernels, scaling up the path extension kernels by a factor of $1.4$ allows the agent to stabilize longer paths.

**Stopping Plan Extension.** Plan extension does not continue indefinitely. It must stop at appropriate boundaries like targets, squares adjacent to boxes, or walls. We observe (Figure 8) that this stopping mechanism is implemented via negative contributions to the path channels at relevant locations. These stopping signals originate from either the encoder or hidden state channels that represent static environmental features (such as those in the 'entity' channel group, Table 8), effectively preventing the plan from extending beyond targets or into obstacles. This aspect of the transition model prevents the DRC from adding impossible transitions to its path.

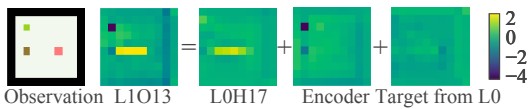

Figure 8: Plan stopping mechanism demonstration shown through $o$-gate contributions of the box-right channel (L1H13). The direct effect shows that convolving the forward and backward right-plan-extension kernel on the converged box-right channel (L0H17) spills into the squares of the box and the target. The encoder and the target channels from layer 0 add a negative contribution to counteract the spillover and stop the plan extension.

**Conclusion** Using the path channel representation allows us to directly examine model weights in order to understand model behavior. We find that the encoder initializes path channel activations near the target and boxes, and extends them through forward and backward path extension kernels. Negative path channel activations prevent the path extension kernels from making impossible movements, serving some of the same features of a transition model.

### 3.3 Pruning path segments

**Backtracking mechanism.** The plan extension kernels serve a dual purpose and also allow the algorithm to backtrack from bad paths. As part of its bidirectional planning, the DRC has forward and backward plan extension kernels, so negative activations at the end of a path are propagated to the beginning by the backward kernel, and negative activations at the beginning of a path are propagated to the end by the forward kernel. This allows the DRC to propagate negative activations along a path, thus pruning unpromising path fragments. See Section I for an example.

**Winner-takes-all mechanism.** To select a single path for a box when multiple options exist, the network employs a Winner-Takes-All (WTA) mechanism among *short-term* path channels. Excluding the *long-term* path channels allows the DRC to maintain plans for later execution without inhibiting them. Figure 10 (bottom-left) shows that weights connecting path channels for various directions cause the path channel activations to inhibit each other at the same square. The direction with the strongest activation suppresses activations in alternative directions which, combined with the sigmoid activation, ensures that only one direction's path channels remain active for imminent execution. We construct a level with equally viable paths to causally demonstrate (Figure 9): initially, both paths have similar activations, but the slightly stronger one quickly dominates in steps 1 and 2 and deactivates the other via this inhibitory interaction. Zero-ablating the kernels between the channels of the two directions eliminates the WTA effect, leaving both potential paths simultaneously active. Thus, we conclude that kernels connecting various short-term box-movement path channels implement this crucial selection mechanism.

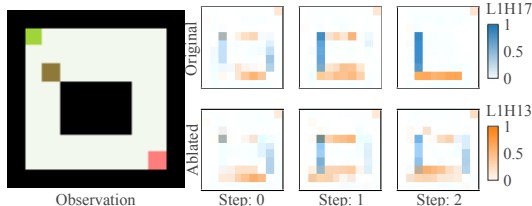

Figure 9: After zero-ablating the kernels connecting the box-down and box-right channels, the WTA mechanism cannot suppress the right-down plan.

In Section N we find the WTA mechanism on four other random training seeds.

### 3.4 Putting it all together

We provide an example where the winner takes all mechanism chooses between two paths, with the plan extension kernels propagating the path selection back to the box and stabilizing the plan in Figure 10.

## 4 Related work

**Mechanistic explanations.** To the best of our knowledge, our work advances the Pareto frontier between complexity of a network and the detail of its characterization, providing the most detailed description of a neural network of this complexity. Much work focuses on the mechanisms of large language models. LLMs are more complex than the DRC, but the algorithms these papers explain are simpler as measured by the size of the abstract causal graph (Geiger et al., 2021; Chan et al., 2022). Examples include work on GPT-2 small (Wang et al., 2023; Hanna et al., 2023; Dunefsky et al., 2024), Gemma 2-2B (Marks et al., 2024; Nanda et al., 2023c), Claude 3.5 Haiku (Lindsey et al., 2025; Marks et al., 2025), and others (Zhou et al., 2024). A possible exception is Lindsey et al. (2025), which contains many simple explanations whose graphs together would add up to a graph larger than that of the present work. However, their explanations rely only on empirical causal effects and are local (only valid in their prompt), contrasting with weight-level analysis that applies to all inputs. Pioneering work in understanding vision models (Olah et al., 2020; Schubert et al., 2021; Voss et al., 2021) is very thorough in labeling features but provides a weight-level explanation for only a small part of InceptionV1 (Cammarata et al., 2021). Other work focuses on tiny toy models and explains their mechanisms very thoroughly, such as in modular addition (Nanda et al., 2023a; Chughtai et al., 2023; Zhong et al., 2023; Quirke & Barez, 2023; Gross et al., 2024; Yip et al., 2025), binary addition (Primozic, 2023), small language transformers (Olsson et al., 2022; Heimersheim & Janiak, 2023), or a transformer that finds paths in small binary trees (Brinkmann et al., 2024).

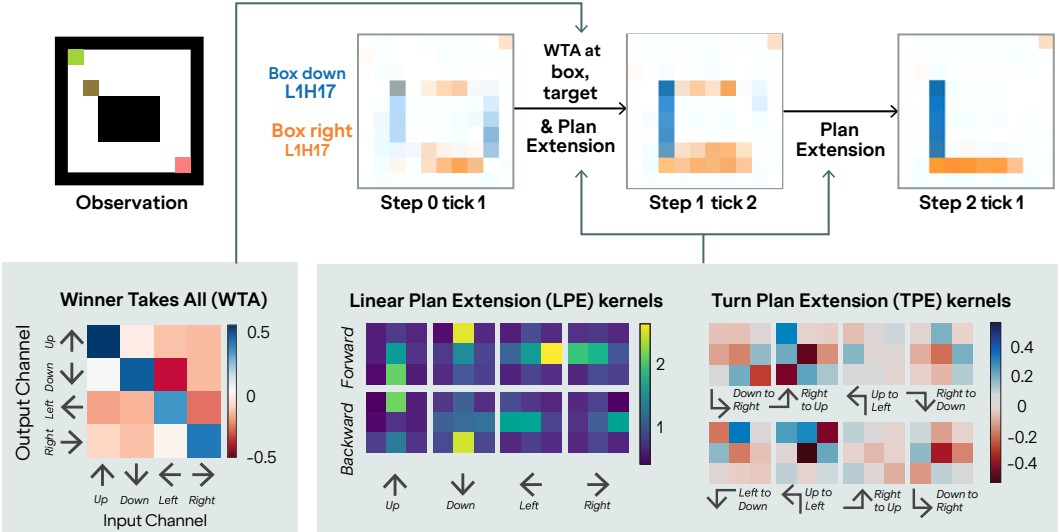

Figure 10: A situation with two equally good paths from the box ■ to the target ■. The sum of box-down (L1H17) and box-right (L1H13) channels shows that the network searches forward from the box and backward from the target. Both paths (down-then-right and right-then-down) are visible at step 0 tick 1 (left) due to the encoder; and the down and right channels have similar activations on the box square (gray). From step 0 tick 1 until step 2 tick 1 (Section 2.1 defines 'tick'), the plans are extended in the same direction by Linear Plan Extension (LPE) kernels (bottom-middle) and extended into switching directions by Turn Plan Extension kernels (bottom-right), stopping (Figure 8) on signals corresponding to reaching the target or hitting obstacles. The plan at the box square is resolved at step 1 tick 2 using a Winner-Takes-All (WTA) mechanism. The average WTA kernel weights (bottom-left figure, averaging over $W_{\cdot h_1}$ and $W_{\cdot h_2}$ for $\cdot \in \{i, j, f, o\}$) subtract each channel from all the others, which through a sigmoid approximates an argmax. The magnitude of the diagonal entries (stronger for down than right) break ties.

**DRC in Sokoban.** Taufeeque et al. (2024); Bush et al. (2025) find internal plan representations in the DRC by predicting future box and agent moves from its activations using logistic regression probes. Some of their probes are causal, others can be used to generalize the DRC to larger levels; however, further analysis is primarily based on qualitative probe and model behavior rather than mechanisms. Our analysis of bidirectional planning is much more mechanistic, and the representation we uncover is much simpler – instead of a probe, it is simply reading off of a channel.

## 5   DISCUSSION AND CONCLUSION

Now we discuss broader takeaways from our analysis of the path representation and construction.

**Probes find a predictive, not causal representation.** Prior probes on the Sokoban network assign weight to both path channels along with channels that are spuriously correlated. The probes thus had lower causal intervention scores compared to the path channels we directly identified.

**Transition model.** Despite training with model-free reinforcement learning, we find some components of a transition model. The plan extension kernels place activation on the tiles that would result from moving in a particular direction. The model also generates negative activation for invalid moves, such as moving into a wall.

**Path channel activations as a value function.** So far, we have discussed the path channel activations as a representation of the path. We believe that the path channel activations also bear similarities to an internal value function, or more precisely a Q function, evaluating whether to take an action at a particular state. Firstly, the plan extension kernels propagate positive and negative activations along path segments to propagate reward information forward and backward. Targets anchor positive

activation for moving in that direction, while obstacles generate negative activation. Negative activation at the end of a path can propagate backwards until it prunes the path segment entirely. Secondly, the winner takes all mechanism uses the path channel activations to choose between conflicting path segments, stabilizing the path to choose (generally) higher activation rewards.

It differs from a typical Q function or informal reward function in several important ways. Firstly, the activations do not appear to correspond to discounted or penalized reward. In the Sokoban environment, each step costs $0.1$ reward, so the reward should decrease over time, with longer paths having a weaker activation. Instead, the activation functions represent the preference for shorter paths implicitly in the dynamics of the planning mechanism. The path extension kernels only extend the path a few moves forward per tick, and so shorter paths to the target generally have their path segments reach boxes before longer path segments. In Section J, we exploit this insight to make the agent take the longer of two paths. Secondly, there are various biases in the winner takes all mechanism. An astute reader might look at Figure 10 and note that the kernel is biased rather than treating all values symmetrically.

**Mesa-optimizers.** Hubinger et al. (2019) introduced the concept of a *mesa-optimizer*, an AI that learns to pursue goals via internal reasoning. Examples of mesa-optimizers did not exist at the time, so subsequent work studied the problem of whether the learned goal could differ from the training signal, *reward misgeneralization* (Di Langosco et al., 2022; Shah et al., 2022). Oswald et al. (2023) argued that transformers do in-context linear regression and are thus mesa-optimizing the linear regression loss, but this doesn't constitute agentic behavior. Modern AI agents appear to reason, but whether they internally optimize a goal is unresolved.

This work answers, in the affirmative, the question of whether or not agentic mesa-optimizers exist. We present a model organism of mesa-optimization, then point to its internal planning process and to its learned value function. The value function differs from what it should be from the training reward, albeit in benign ways: the training reward has a $-0.1$ per-step term, but the value encoded in the path channels *do not* capture plan length at all. In fact, which path the DRC picks is a function of which one connects to the target first, encoding the preference for shorter paths purely in the LPE and TPE kernels (Section J). To compute the value head (critic), the DRC likely counts how many squares are active in the path channels.

**Conclusion** In conclusion, we discover a simple representation of the DRC$(3, 3)$ agent's intended path as activations in its *path channels*. These path channels are initialized by encoder kernels, then extended bidirectionally by forward and backward plan extension kernels, and stabilized by a winner takes all kernel. The agent is able to plan forward and backtrack by increasing or decreasing path channel activations via the same plan extension kernels.

## LLM Usage Statement

LLMs were used for basic writing feedback and title brainstorming, but not direct contribution.

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

# Appendix

## A    COMMON COMPONENTS OF SEARCH ALGORITHMNS

A search algorithm requires four key components:

1. A representation of states.
2. A transition model that defines which nodes (states) are reachable from a currently expanded node when taking a certain action.
3. A heuristic function that determines which nodes to expand.
4. A value function that determines which plan to choose once the search ends (in online search algorithms, only the first action from the chosen plan is taken).

The heuristic varies by algorithm:

- For A*, it is distance$(n)$ + heuristic$(n)$. (Russell & Norvig, 2009)
- For iterative-deepening alpha-beta search (as used in Stockfish), the heuristic comprises move ordering and pruning criteria. (Chess Programming Wiki, 2024)
- For AlphaZero/MuZero MCTS, it uses the UCT formula pre-rollout, incorporating backed-up value functions and a policy with Dirichlet noise. (Silver et al., 2018; Schrittwieser et al., 2019)

In all cases, the expansion process influences the relative evaluation of actions in the starting state. The final action selection relies on a value function:

- A*: Uses the actual path distance when plans have been fully expanded. (Russell & Norvig, 2009)
- AlphaZero/MuZero MCTS: Employs backpropagated estimated values combining rollout and final score. (Silver et al., 2018; Schrittwieser et al., 2019)
- Stockfish 16+: Utilizes the machine-learned evaluation function at leaf nodes. (Chess Programming Wiki, 2024)

In the body of the paper, we show how various parts of the trained DRC correspond to these four components.

## B    NETWORK ARCHITECTURE

The DRC architecture consists of an convolutional encoder $E$ without any non-linearities, followed by $D$ ConvLSTM layers that are repeated $N$ times per environment step, and an MLP block that maps the final layer's hidden state to the value function and action policy.

The encoder maps the observation $x_t$ at timestep $t$ to the encoded state $e_t$. For all $d > 1$, the ConvLSTM layer updates the hidden state $h_d^n, c_d^n$ at each tick $n$ using the following equations:

$$e_t := E(x_t) = W_{E_2} * (W_{E_1} * x_t + b_{E_1}) + b_{E_2} \tag{3}$$

$$c_d^n, h_d^n := \text{ConvLSTM}_d(e_t, h_{d-1}^n, c_d^{n-1}, h_d^{n-1}) \tag{4}$$

$$i_d^n := \tanh(W_{ie}^d * e_t + W_{ih_1}^d * h_{d-1}^n + W_{ih_2}^d * h_d^{n-1} + b_i) \tag{5}$$

$$f_d^n := \sigma(W_{fe}^d * e_t + W_{fh_1}^d * h_{d-1}^n + W_{fh_2}^d * h_d^{n-1} + b_f) \tag{6}$$

$$j_d^n := \sigma(W_{je}^d * e_t + W_{jh_1}^d * h_{d-1}^n + W_{jh_2}^d * h_d^{n-1} + b_j) \tag{7}$$

$$c_d^n := f_d^n \odot c_d^{n-1} + i_d^n \odot j_d^n \tag{8}$$

$$o_d^n := \tanh(W_{oe}^d * e_t + W_{oh_1}^d * h_{d-1}^n + W_{oh_2}^d * h_d^{n-1} + b_o) \tag{9}$$

$$h_d^n := o_d^n \odot \tanh(c_d^n) \tag{10}$$

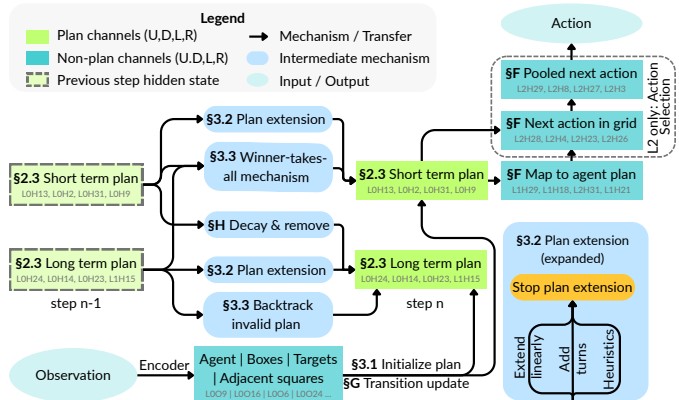

Figure 11: The planning algorithm circuit learned by $\mathrm{DRC}(3,3)$. While the plan nodes are present and updated across all the layers, this circuit only shows the short and long-term plan nodes in the first layer's hidden state (L0HX) with a channel X for each direction up, down, left, and right. Mechanisms are annotated with the sub-section they are studied in Section 3.

Here $*$ denotes the convolution operator, and $\odot$ denotes point-wise multiplication. Note that $\theta_d = (W_{i\cdot}, W_{j\cdot}, W_{f\cdot}, W_{o\cdot}, b_i, b_j, b_f, b_o)_d$ parameterizes the computation of the $i, j, f, o$ gates. For the first ConvLSTM layer, the hidden state of the final ConvLSTM layer is used as the previous layer's hidden state.

A linear combination of the mean- and max-pooled ConvLSTM activations is injected into the next step, enabling quick communication across the receptive field, known as pool-and-inject. A boundary feature channel with ones at the boundary of the input and zeros inside is also appended to the input. These are ignored in the above equations for brevity.

Finally, an MLP with 256 hidden units transforms the flattened ConvLSTM outputs $h_D^N$ into the policy (actor) and value function (critic) heads. In our setup, $D = N = 3$ and $C = 32$ matching Guez et al. (2019)'s original hyperparameters. An illustration of the full architecture is shown in Figure 2.

## C  NETWORK TRAINING DETAILS

The network was trained using the IMPALA V-trace actor-critic (Espeholt et al., 2018) reinforcement learning (RL) algorithm for $2 \cdot 10^9$ environment steps with Guez et al.'s Deep Repeating ConvLSTM (DRC) recurrent architecture consisting of three layers repeated three times per environment step, as shown in Figure 2.

The observations are $H \times W$ RGB images with height $H$ and width $W$. The agent, boxes, and targets are represented by the green ■, brown ■, and red ■ pixels respectively (Schrader, 2018), as illustrated in Figure 12. The environment has -0.1 reward per step, +10 for solving a level, +1 for putting a box on a target and -1 for removing it.

**Dataset**  The network was trained on 900k levels from the unfiltered train set of the Boxoban dataset (Guez et al., 2018). Boxoban separates levels into train, validation, and test sets with three difficulty levels: unfiltered, medium, and hard. The hard set is a single set with no splitting. Guez et al. (2019) generated these sets by filtering levels unsolvable by progressively better-trained DRC networks. So easier sets occasionally contain difficult levels. Each level in Boxoban has 4 boxes in a grid size

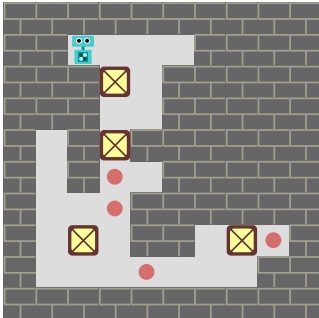 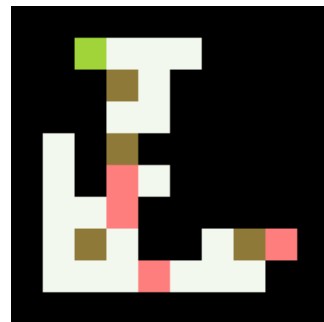

Figure 12: High resolution visualization of a Sokoban level along with the corresponding symbolic representation that the network observes. The agent, boxes, and targets are represented by the green ■, brown ■, and red ■ squares respectively.

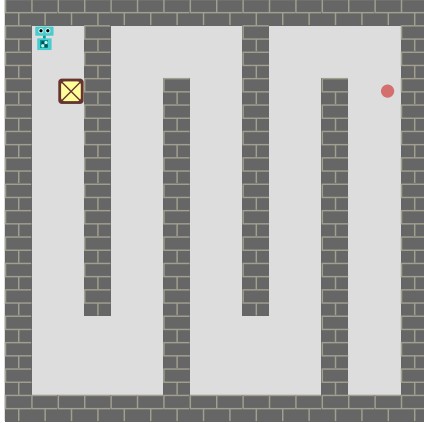

Figure 13: $16 \times 16$ zig-zag level that the original DRC$(3,3)$ network fails to solve. Steering $W_{ch1}^d$ and $W_{ch_2}^d$ by a factor of $1.2$ solves this level and similar zig-zag levels for sizes upto $25 \times 25$.

of $H = W = 10$. The $H \times W$ observations are normalized by dividing each pixel component by 255. The edge tiles in the levels from the dataset are always walls, so the playable area is $8 \times 8$. The player has four actions available to move in cardinal directions (Up, Down, Left, Right). The reward is -0.1 per step, +1 for placing a box on a target, -1 for removing it, and +10 for finishing the level by placing all of the boxes. In this paper, we evaluate the network on the validation-medium and hard sets of the Boxoban dataset. We also often evaluate the network on custom levels with different grid sizes and number of boxes to clearly demonstrate certain mechanisms in isolation.

**Action probe for evaluation on larger grid sizes**   The DRC$(3,3)$network is trained on a fixed $H \times W$ grid size with the hidden state channels flattened to a $H \times W \times C$ tensor before passing it to the MLP layer for predicting action. Due to this limitation, the network cannot be directly evaluated on larger grid sizes. Taufeeque et al. (2024) trained a probe using logitic regression with 135 parameters on the hidden state $h$ of the final ConvLSTM layer to predict the next action. They found that the probe can replace the 0.8M parameter MLP layer to predict the next action with a 77.9% accuracy. They used this probe to show that the algorithm learned by the DRC backbone generalizes to grid sizes 2-3 times larger in area than the training grid size of $10 \times 10$. We use these action probes to run the same network on larger grid sizes in this paper.

## D   GATE IMPORTANCE

We identify here the components that are important and others which can be ignored. We noticed that our analysis can be simplified by ignoring components like the previous cell-state $c$ and forget gate $f$ that don't have much effect. On mean-ablating the cell-state $c$ at the first tick $n = 0$ of every

Table 3: Comparison of network intervened with single-step cache across different channel groups. We report the percentage drop of solve rate compared to the original network (%) on medium-difficulty levels.

| Group | # Channels | Performance Drop |
|---|---|---|
| Non-planning | 37 | $10.5 \pm 1.9$ |
| Planning | 59 | $57.6 \pm 2.8$ |
| Random planning subset | 37 | $41.3 \pm 2.4$ |

step for all the layers, we find that the network's performance drops by $21.28\% \pm 0.04\%$. The same ablation on the forget gate $f$ results in a drop of $2.66\% \pm 0.03\%$. On the other hand, the same ablation procedure on any of the other gates $i$, $j$, $o$, or the hidden state $h$ breaks the network and results in a drop of $100.00\%$ with no levels solved at all. This shows that the forget gate is not as important as other gates in regulating the information in the cell-state, and the information in the cell-state itself is not relevant for solving most levels. The only place we found the forget gates to be important is for accumulating the next-action in the GNA channels (Section F).

The mean-ablation experiment shows that the network computation from previous to the current step can be simplified to the following:

$$c_d^n \approx E[f_d^n] \odot E[c_d^{n-1}] + i_d^n \odot j_d^n = \mu + i_d^n \odot j_d^n \tag{11}$$
$$h_d^n = o_d^n \odot \tanh(c_d^n) \approx o_d^n \odot \tanh(\mu + i_d^n \odot j_d^n) \tag{12}$$

We therefore focus more on the $i, j, o$ gates and the hidden state $h$ in our analysis in this paper. Qualitatively, it also looks like the cell-state $c$ is very similar to the hidden state $h$. Note that the cell state $c$ not being much relevant doesn't imply that the network is not using information from previous hidden states, since most of the information from the previous hidden states $h_d^{n-1}$ flows through the $W_{ch_2}^d$ kernels.

## E    LABEL VERIFICATION AND OFFSET COMPUTATION

We see from Table 9 that most channels can be represented with some combination of features that can be derived from observation image (base feature) and future box or agent movements (future features). We compute the following 5 base features: agent, floor, boxes not on target, boxes on target, and empty targets. For future features, we get 3 features for each direction: box-movement, agent-movement, and a next-action feature that activates positively on all squares if that action is taken by the network at the current step. We perform a linear regression on the 5 base and 12 future features to predict the activations of each channel in the hidden state $h$.

**Offset computation**    On visualizing the channels of the $DRC(3,3)$ network, we found that the channels are not aligned with the actual layout of the level. The channels are spatially-offset by a few squares in the cardinal directions. To automatically compute the offsets, we perform linear regression on the base and future features to predict the channel activation by shifting the features along $x, y \in \{-2, -1, 0, 1, 2\}$ and selecting the offset regression model with the lowest loss. The channels offsets are available in Table 5. We manually inspected all the channels and the offset and found that this approach accurately produces the correct offset for all the 96 channels in the network. All channel visualization in the paper are shown after correcting the offset.

**Correlation**    The correlation between the predicted and actual activations of the channels is provided in the Tables 6 and 7. We find that box-movement, agent-movement, combined-plan, and target channels have a correlation of $66.4\%$, $50.8\%$, $48.0\%$, and $76.7\%$. As expected, the unlabeled channels do not align with our feature set and have the lowest correlation of $40.2\%$. Crucially, a baseline regression using only base features yielded correlations below $20\%$ for all channels, confirming that the channels are indeed computing plans using future movement directions. These correlations should be treated as lower bounds, as this simple linear approach on the binary features cannot capture

many activation dynamics like continuous development, representation of rejected alternative plans (Section 3.3), or the distinct encoding of short- vs. long-term plans.

# F  PLAN REPRESENTATION TO ACTION POLICY

The plan formed by the box movement channels are transferred to the agent movement channels. For example, Figure 23b shows that the agent down movement channel L1H18 copies the box down movement channel L1H17 by shifting it one square up, corresponding to where the agent will push the box. The kernels also help in picking a single path if the box can go down through multiple paths.

Once the box-plan transfers to the agent-movement channels, these channels are involved in their own agent-path extension mechanism. Figure 23a show that the agent-movement channels have their own linear-plan-extension kernels. These channels also have stopping conditions that stop the plan-extension at the box squares and agent square. Thus, as a whole, the box-movement channels find box to target paths and the agent-movement channels copy those paths and also find agent to box paths.

Finally, the network needs to find the next action to take from the complete agent action plan represented in agent-movement channels. We find that the network dedicates separate channels that extract the next agent action. We term these channels as the grid-next-action (GNA) channels (Table 8). There exists one GNA channel for each of the four action directions. A max-pooling operation on these channels transfers the high activation of an action to the entire grid of the corresponding agent action channel. We term these as the pooled-next-action (PNA) channels (Table 8). Lastly, the MLP layer aggregates the flattened neurons of the PNA channels to predict the next action. We verify that the PNA and GNA channels are completely responsible for predicting the next action by performing causal intervention that edits the activation of the channel based on our understanding to cause the agent to take a random action at any step in a level. Table 2 shows that both the PNA and GNA channels are highly accurate in modifying the next action. We now describe how the network extracts only the next agent move into the GNA channels.

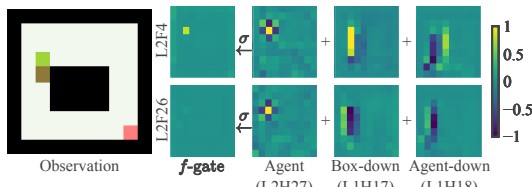

The individual gates of the GNA channels copy activations of the agent-movement channels. Some gates perform subtraction of the agent and box movement channels to get agent-exclusive moves and the next agent box push. Figure 14 (top-right) shows one such example where the agent and box movement channels from layer 1 are subtracted resulting in an activation exclusively at the agent square. The GNA gates also receive positive activation on the agent square through L2H27 which detects agent at the first tick $n = 0$ of a step. Figure 14 shows that the $f$-gate of all GNA channels receives a positive contribution from the agent square. To counteract this, the agent-movement channels of one direction contribute negatively to the GNA channels of all other directions. All of this results in the agent square of the GNA channel of the next move activating strongly at the second tick $n = 1$.

Figure 14: **Left:** Observation at step 3 where the agent moves down.**Right:** The GNA channels, which represent the direction that the agent will move in at the next step, predict the agent moving down primarily through $f$-gate. The box- and agent-down channels are offset and subtracted to get the action at the agent square. The checkered agent location pattern from L2H27 also helps in isolating the action on the agent square. The active $f$-gate square accumulates activation in the cell-state $c$ which after max-pooling and MLP layer decodes to the down action being performed.

Thus we have shown that the complete plan is filtered through the GNA channels to extract the next action which activates the PNA channel for the next action to be taken.

# G  STATE TRANSITION UPDATE

We have understood how the plan representation is formed and mapped to the next action to be taken. However, once an action is taken, the network needs to update the plan representation to reflect the new state of the world. We saw in Figure 5 that the plan representation is updated by

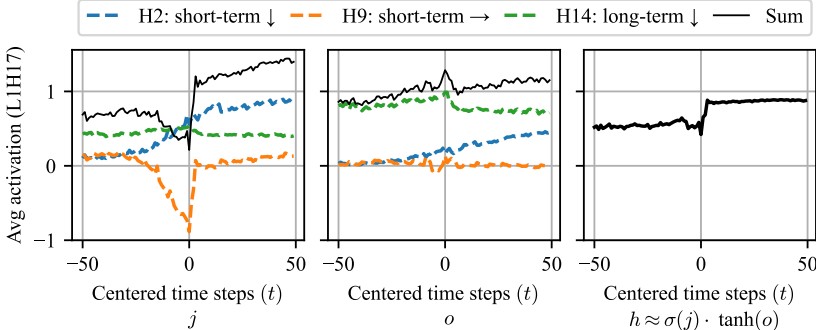

Figure 15: Transfer mechanism from long to short-term channel shown through contributions into the gates of the short-term-down (L1H17) channel averaged across squares where a right box-push happens at $t = 0$ and down box-push later on. The long-term-down channel L0H14 contributes to the $o$-gate at all steps $t$. However, L0H9 (short-term-right) activates negatively in the sigmoid $j$-gate, thus deactivating L1H17. As the right move gets played at $t = 0$, L0H9's negative contribution vanishes, enabling the transfer of L0H14 and L0H2 into L1H17.

deactivating the square that represented the last action in the plan. This allowed a different future action to be represented at the same square in the short-term channel which was earlier stored only in the long-term channel. We now show how a square is deactivated in the plan representation.

After an action is taken, the network receives the updated observation on the first tick $n = 0$ with the new agent or box positions. The combined $W_{ce}^d$ kernels for each layer that map to the path channels contain filters that detect only the agent, box, or target, often with the opposite sign of activation of the plan in the channel (Figure 24). Hence, when the observation updates with the agent in a new position, the agent kernels activates with the opposite sign of the plan activation that deletes the last move from the plan activation in the hidden state. The activation contributions in Figure 8 shows the negative contribution from the encoder kernels on the agent and the square to the left of the box. Therefore, the agent and the boxes moving through the level iteratively remove squares from the plan when they are executed with the plan-stopping mechanism ensuring that the plan doesn't over-extend beyond the new positions from the latest observation.

## H ACTIVATION TRANSFER MECHANISM BETWEEN LONG AND SHORT TERM CHANNELS

Consider a scenario where two different actions, $A_1$ and $A_2$ ($A_1 \neq A_2$), are planned for the same location ("square") at different timesteps, $t_1$ and $t_2$, with $t_1 < t_2$. As illustrated in Section 3.3 and further detailed in Figure 5, the later action ($A_2$ at $t_2$) is initially stored in the long-term channel for timesteps $t < t_1$. This information is transferred to the short-term channel only after the earlier action ($A_1$) is executed at $t = t_1$. We now describe the specific mechanism responsible for this transfer of activation from the long-term to the short-term channel.

In Figure 15, the activations transfer into L1H17 (short-term-down) from L0H14 (long-term-down) and L0H2 (short-term-down) channels when a right action is taken at $t = 0$ represented in L0H9 (short-term-right). The short-term-right channel L0H9 imposes a large negative contribution via the $j$-gate to inhibit L1H17, keeping it inactive even as the long-term-down channel tries to transfer a signal through the $j$ and $o$-gates for $t < 0$. Once the first move completes ($t = 0$), short-term-right is no longer active and so the inhibition ceases. The removal of the negative input allows the $j$-gate's activation to rise, enabling the long-term-down activation transfer through $o$-gate, making it the new active short-term action at the square. This demonstrates how long-term channels hold future plans, insulated from immediate execution conflicts by the winner takes all (WTA) mechanism (Section 3.3 and Figure 10) acting on short-term channels.

# I  CASE STUDY: BACKTRACKING MECHANISM

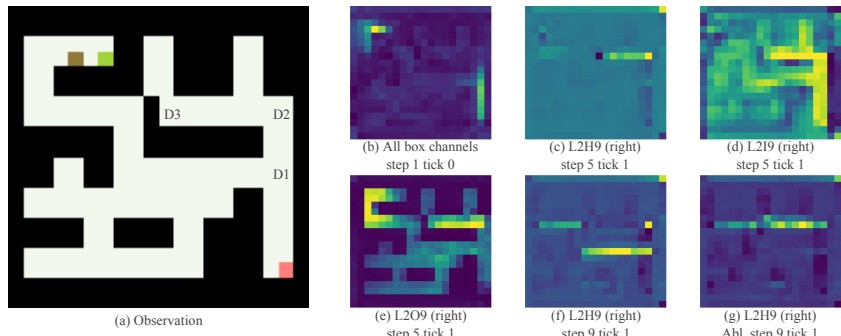

Figure 16: **(a)** $20 \times 20$ level we term as the "backtrack level" with key decision nodes D1-D3 for backward chaining. **(b)** The sum of box-movement channels at step 1 tick 0 indicates forward (from box) and backward (from target) chaining. **(c-e)** Activation of the box-right channel L2H9 involved in backward chaining at step 5 tick 1. Backward chaining moved up from D1 to D2 and then hitting a wall at D3, which initiates backtracking towards D2 through negative plan extension. The negative wall activation comes from the $o$-gate of L2H9. **(f)** Successful pathfinding at T28 after backtracking redirected the search. **(g)** Ablation: Forcing positive activation at D3 (by setting it to its absolute value) prevents backtracking, hindering correct solution finding (L2H9 Abl., T28).

In particular, the forced positive ablation at D3 results in an incorrect plan (g) which seemingly goes right all the way through the wall, as opposed to the correct plan (f) which goes right on a valid path.

Consider the level depicted in Figure 16 (a). The network begins by chaining forward from the box and backward from the target(Figure 16, b). Upon reaching the square marked D1, the plan can continue upwards or turn left. Here, the linear and turn plan-extension kernels activate the box-down and the box-right channels, respectively. However, the box-down activation is much higher because the weights of the linear extension kernels are much larger than the turn kernels (as seen in Figure 10). Due to this, the winner-takes-all mechanism leads to the search continuing upwards in the box-down channel. Upon hitting a wall at D2, the chain turns right along the 'box-right channel' (L2H9) and continues until it collides with another wall at D3. (Figure 16, c).

This triggers backtracking. While both $i$-gate and $o$-gate activations contribute to plan extension, the $o$-gate also activates strongly *negatively* on wall squares like D3 (Figure 16d, e). This leads to a dominant negative activation in the 'box-right' channel, which then propagates backward along the explored path (from D3 towards D2) via the forward plan-extension kernels of L2H9.

This weakens the dominant 'box-down' activation at D1, allowing the alternative 'box-right' path from D1 to activate. The search then proceeds along this new route, allowing the backward chain to connect with the forward chain, resulting in the correct solution (Figure 16, f).

To verify this mechanism, we performed an intevention by forcing the activation at the wall squares near D3 to be positive (by taking their absolute values). This blocked backtracking, and the network incorrectly attempted to connect the chains through the wall (Figure 16, g). This confirms that negative activation generated at obstacles is the key driver for backtracking, and is what allows the network to discard failed paths and explore alternatives. We quantitatively test this claim further by performing the same intervention on transitions from 512 levels where a plan's activation is reduced by more than half in a single step which was preceded by a neighboring square having negative activation in the path channel. We define the intervention successful if forcing the negative square to an absolute value doesn't reduce the activation of the adjacent plan square. The intervention results in a success rate with $95\%$ confidence intervals of $85.1\% \pm 5.0\%$ and $48.9\% \pm 3.3\%$ for long- and short-term channels, respectively. This checks out with the fact that long-term channels represent plans not in the immediate future which would get backtracked through negative path activations. On the other hand, negative activations in the short-term channels are also useful during the winner-takes-all (WTA) mechanism and deadlock prevention heuristics. Filtering such activations for short-term channels from the intervention dataset would improve the numbers.

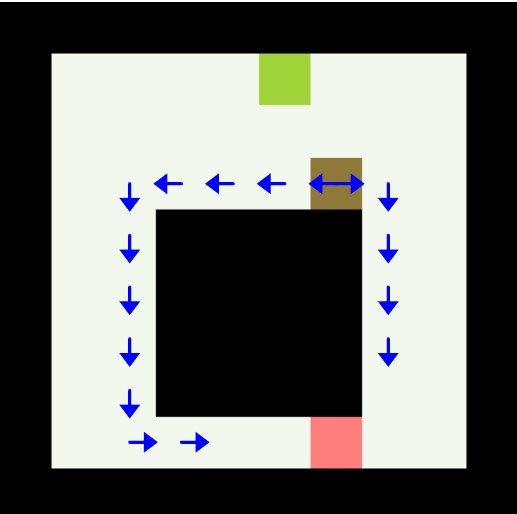

Figure 17: A level with two paths, one longer than the other. We initialize the starting hidden state with the two paths shown such that they both have two squares left to reach the target. We find that the expands both paths and picks the left (longer) path through the winner-takes-all mechanism since it reaches there with higher activation through linear-plan-extension.

## J  CASE STUDY: MAKING THE NETWORK TAKE THE LONGER PATH

The network usually computes the shortest paths from a box to a target by forward (from box) and backward (from target) chaining linear segments until they connect at some square as illustrated in Figure 10. As soon as a valid plan is found for a box along one direction, the winner-takes-all mechanism stabilizes that plan through its stronger activations and deletes any other plans being searched for the box. From this observation, we hypothesize that the network values finding valid plans in least number of steps than picking the shorter one. We verify this value preference of the network by testing the network with on the level shown in Figure 17 with the starting state initialized with the two paths shown. The left path (length=13) is longer than the right path (length=7) for reaching from the box to target. Both paths are initialized in the starting hidden state to have two arrow left to complete the path. We find that in this case, both the paths reach the target, but the left one is stronger due to linear plan extension kernels reaching with higher activation. This makes the network pick the left path and prune out the shorter right path. If we modify the starting state such that left and right paths have 3 and 2 square left to the reach the target, then the right path wins and the left path is pruned out. This confirms that the network's true value in this case is to pick a valid plan closer to target than to pick a shorter plan. However, since convolution moves plan one square per operation, the network usually seems to have the value of picking the shorter plan.

## K  CHANNEL REDUNDANCY

We see from Table 8 that the network represents many channels per box-movement and agent-movement direction. We find at least two reasons for why this redundancy is useful.

First, it facilitates faster spatial propagation of the plan. Since the network uses $3 \times 3$ kernels in the ConvLSTM block, information can only move 1 square in each direction per convolution operation. By using redundant channels across multiple layers, the network can effectively move plan information several squares within a single time step's forward pass (one square per relevant layer). Evidence for this rapid propagation is visible in Figure 16(b), where plan activations extend 7-10 squares from from the target and the box within the first four steps on a $20 \times 20$ level.

Second, the network dedicates separate channels to represent the plan at different time horizons. We identified distinct short-term (approximately 0-10 steps ahead) and long-term (approximately 10-50 steps ahead) channels within the box and agent-movement categories.

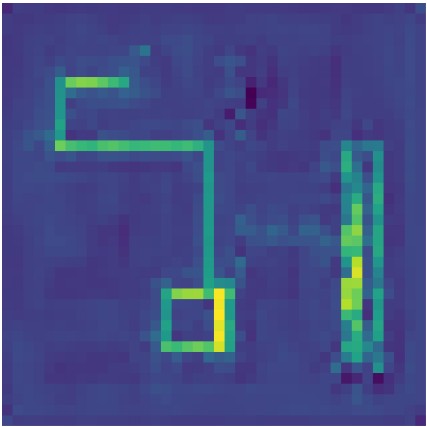

Figure 18: Sum of activations of box-movement channels on the $40 \times 40$ backtrack level with the network weights $W_{ch1}^d$ and $W_{ch_2}^d$ steered by a factor of 1.4. The planning representation gets stuck in the loop shown, unable to backtrack and explore other paths. The activations of other squares become chaotic, changing rapidly and randomly on each step.

This allows the network to handle scenarios requiring the same location to be traversed at different future times. For example, if a box must pass through the same square at time $t_1$ and later at time $t_2$, the network can use the short-term channel to represent the first push at $t_1$ and the long-term channel to represent the second push at $t_2$. Figure 5 (right) illustrates this concept, showing activation transferring from a long-term to a short-term box-down-movement channel once the earlier action at that square is taken by the agent.

## L   WEIGHT STEERING FIXES FAILURE ON LARGER LEVELS

Previous work (Taufeeque et al., 2024) showed that, although the DRC$(3, 3)$ network can solve much bigger levels than $10 \times 10$ grid size on which it was trained, it is easy to contruct simple and natural adversarial examples which the network fails to solve. For example, the $n \times n$ zig-zag level in Figure 13 that can be scaled arbitrarily by adding more alleys and making them longer, is only solved for $n \leq 15$ and fails on all $n > 15$. The big level shown in Figure 16 (a) is solved by the network on the $20 \times 20$ grid size but fails on $30 \times 30$ or $40 \times 40$ grid size.

Figure 22 (a) visualizes the sum of activations of the box-movement channels on a $40 \times 40$ variant of the backtrack level in which we see the reason why larger levels fail: the channel activations decay as the plan gets extended further and further. This makes sense as the network only saw $10 \times 10$ levels during training and hence the kernel weights were learned to only be strong enough to solve levels where targets and boxes are not too far apart. We find that multiplying the weights of $W_{ch1}^d$ and $W_{ch_2}^d$, the kernels that update and maintain the hidden state, by a factor of 1.2 helps the network extend the plan further. This weight steering procedure is able to solve the zig-zag levels for sizes up to $n = 25$ and the backtrack level for sizes up to $40 \times 40$. Figure 22 (b, c) show that upon weight steering, the box-movement channels are able to maintain their activations for longer, enabling the network to solve the level. However, for much larger levels, weightsteered networks also fall into the same trap of decaying activations, failing to extend the plan. Further weight steering with a larger factor can help but we find that it can become brittle, as the planning representation gets stuck in wrong paths, unable to backtrack, with the activations becoming chaotic (Figure 18). We also tried other weight steering approaches such as multiplying all the weights of the network by a factor or a subset such as the kernels of path channels, but find that they do not work as well as the weight steering of $W_{ch1}^d$ and $W_{ch_2}^d$.

## M   EMERGENCE OF PLANNING STRUCTURE DURING TRAINING

To understand when the planning mechanisms described in this paper emerge during training, we analyzed intermediate checkpoints saved throughout the 2 billion environment steps of training.

Specifically, we tracked the evolution of the Winner-Takes-All (WTA) mechanism by measuring kernel connectivity between direction channels.

**Methodology.** We analyzed the convolutional kernels connecting short-term box-movement path channels across all available checkpoints. For each checkpoint, we computed:

1. **Self-connection strength**: The average weight magnitude of kernels connecting each direction channel to itself (e.g., box-down $\rightarrow$ box-down). Positive values indicate self-reinforcement.

2. **Cross-inhibition strength**: The average weight magnitude of kernels connecting different direction channels (e.g., box-down $\rightarrow$ box-right). Negative values indicate mutual inhibition.

3. **Normalized Self-Cross Difference**: A metric bounded between $-1$ and $+1$, computed as:

$$\text{WTA}_{\text{norm}} = \frac{\text{self} - \text{cross}}{|\text{self}| + |\text{cross}|} \tag{13}$$

A value of $+1$ indicates ideal WTA structure (positive self-connection, negative cross-inhibition), while $-1$ indicates the opposite pattern.

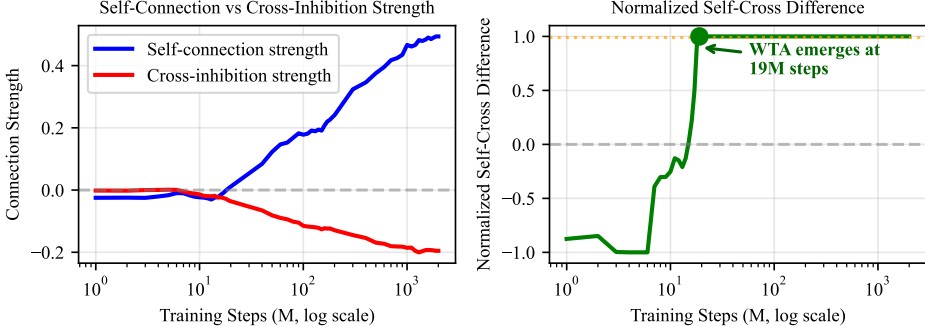

Figure 19: Emergence of the Winner-Takes-All mechanism during training. **Left:** Self-connection strength (blue) increases while cross-inhibition strength (red) becomes increasingly negative over training. **Right:** The normalized self-cross difference transitions sharply from $-1$ to $+1$ around 19M environment steps, indicating rapid emergence of WTA structure.

**Results.** Figure 19 shows the evolution of these metrics across training and shows that plan extension kernels and WTA kernels emerge in distinct phases during training.

1. **Early training (0–10M steps):** Both self-connection and cross-inhibition weights remain near zero. The normalized difference is approximately $-1$, indicating no WTA structure—if anything, the opposite pattern (cross-excitation).

2. **Transition phase (10–19M steps):** A rapid phase transition occurs where self-connection becomes positive while cross-inhibition becomes negative.

3. **WTA emergence (19M steps):** The normalized self-cross difference reaches $+1$, indicating that the full WTA structure has emerged. This occurs at less than 1% of total training (19M of 2000M steps).

4. **Continued refinement (19M–2000M steps):** After the WTA structure emerges, the absolute magnitudes continue to increase (self-connection reaches $\sim 0.5$, cross-inhibition reaches $\sim -0.2$), but the qualitative structure remains stable.

This analysis complements our weight-level mechanistic findings by showing that the planning structures we identify in the fully-trained network emerge through a distinct developmental trajectory during learning.

# N    STABLE PLANNING ALGORITHM ACROSS TRAINING SEEDS

We also found long- and short-term path channels, plan extension kernels, and a winner takes all kernel in four additional random training seeds for the DRC$(3, 3)$ agent, corroborating our overall findings. To do so, we created a simple automated method to label long- and short-term path channels, then looked at their aggregated kernels using the same methodologies as in Figure 7b and Figure 10.

**Automatic discovery method.**    For each channel, we compute the AUC score for predicting future box or agent movements in a each direction, searching over different spatial offsets and activation signs. We assign a given channel to a label if its AUC for some property is over 0.95. For grid-next-action channels we use the channels in the middle tick of the DRC to compute the AUC, and for all others we use the last tick of the DRC.

- *Long-term path channels* are based on the AUC of predicting an action is between 10 and 50 timesteps, and if the AUC for predicting at 50 timesteps out is higher than its AUC for predicting the next timestep.
- *Short-term path channels* are based on the AUC for predicting an action in the next 10 steps, and if the AUC for predicting the next timestep is higher than its AUC for predicting 50 timesteps out.
- *Grid-next-action (GNA) channels* are based on the AUC of predicting just the next move, and not being a short-term path channel.
- *Pooled-next-action (PNA) channels* are based on the AUC of predicting the next move by mean-pooling the activations across spatial dimensions.

We find that the automatic discovery method correctly labels 3/4 of the GNA channels, all the PNA channels, and discovers the box or agent channels with a F1-score of 73.7%.

**Channel statistics.**    Table 4 shows the discovered channel counts across all five networks. Despite independent training, all networks develop qualitatively similar structure: approximately 27 box-movement channels, 7 agent-movement channels, and 3–4 GNA/PNA channels each. All networks also develop both short-term and long-term plan representations.

Table 4: Discovered planning channels across five independently trained networks.

| Seed | Total | Box | Agent | Long-term | Short-term | GNA | PNA |
|------|-------|-----|-------|-----------|------------|-----|-----|
| bkynosqi (Manual) | 38 | 31 | 7 | 5 | 33 | 3 | 4 |
| gobfm3wm | 28 | 21 | 7 | 2 | 26 | 3 | 3 |
| jl6bq8ih | 36 | 28 | 8 | 5 | 31 | 4 | 4 |
| q4mjldyy | 35 | 27 | 8 | 5 | 30 | 4 | 3 |
| qqp0kn15 | 33 | 27 | 6 | 5 | 28 | 3 | 3 |
| Mean | 34.0 | 26.8 | 7.2 | 4.4 | 29.6 | 3.4 | 3.4 |

**Plan extension kernels.**    Figure 20 shows the averaged plan extension kernels for box-movement channels across all networks. Each network develops the characteristic forward and backward propagation pattern: high weights in the direction of plan extension and near-zero weights elsewhere. This confirms that the linear plan extension mechanism emerges consistently across training runs.

**Winner-takes-all structure.**    Figure 21 shows the WTA connectivity matrices for all networks. Each matrix shows the average kernel weight connecting box-movement channels of one direction to another. Most networks exhibit the characteristic WTA pattern: positive self-connections along the diagonal (self-reinforcement) and negative or weaker off-diagonal connections (cross-inhibition between competing directions).

These results support the claim that the path channels and extension kernels discovered in this paper represent a stable solution for DRC(3, 3) that emerges consistently across independent training runs.

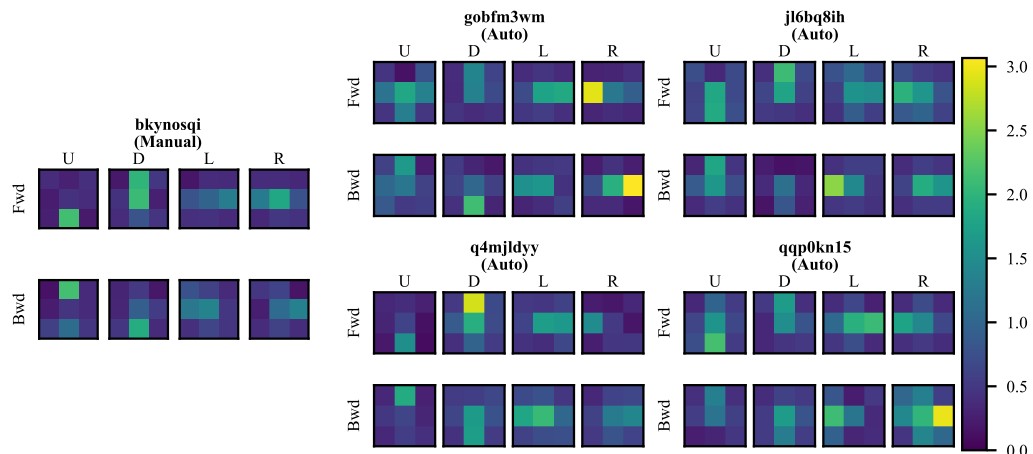

Figure 20: Plan extension kernels across four networks trained independently with different seeds. Top row: forward propagation. Bottom row: backward propagation. Each column group shows the four directions (U, D, L, R) for one network. All networks develop similar linear extension patterns. The unique IDs correspond to the IDs of training run. Manual was computed using our manual channel labeling from the main network in the paper.

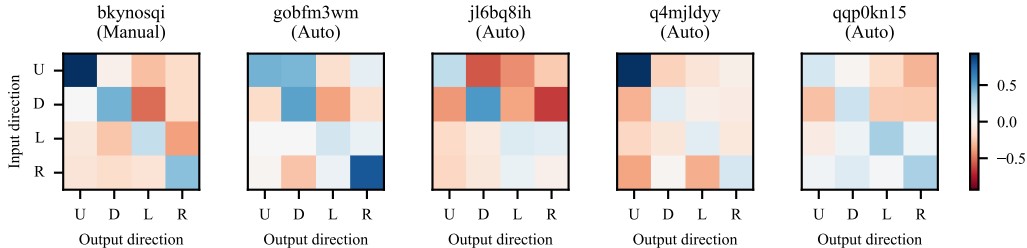

Figure 21: Winner-takes-all connectivity across four networks trained independently with different seeds. Blue indicates positive (reinforcing) connections; red indicates negative (inhibiting) connections. All networks develop self-reinforcement on the diagonal and cross-direction-inhibition off-diagonal. The unique IDs correspond to the IDs of training run. Manual was computed using our manual channel labeling from the main network in the paper.

## O    RELATED WORK

**Mechanistic interpretability.**    Linear probing and PCA have been widely successful in finding representations of spatial information (Wijmans et al., 2023) or state representations and game-specific concepts in games like Maze (Ivanitskiy et al., 2023; Knutson et al., 2024; Mini et al., 2023), Othello (Li et al., 2023; Nanda et al., 2023b), and chess (McGrath et al., 2021; Schut et al., 2023; Karvonen, 2024). However, these works are limited to input feature attribution and concept representation, and do not analyze the algorithm learned by the network. Recent work has sought to go beyond representations and understand key circuits in agents. It is inspired by earlier work in convolutional image models (Carter et al., 2019) discovering the circuits responsible for computing key features like edges, curves, and spatial frequency (Cammarata et al., 2021; Olah et al., 2020; Schubert et al., 2021). In particular, recent work has found mechanistic evidence for few-step lookahead in superhuman chess networks (Jenner et al., 2024; Schut et al., 2023), and future token predictions in LLMs on tasks like poetry and simple block stacking (Lindsey et al., 2025; Men et al., 2024; Pal et al., 2023). However, these works still focus on particular mechanisms in the network rather than a comprehensive understanding of the learned algorithm.

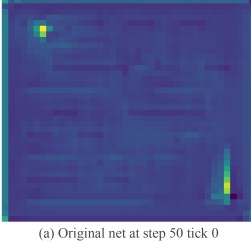 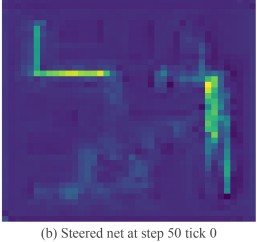 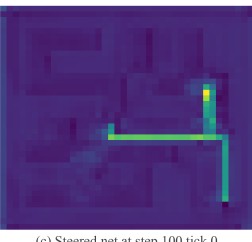

(a) Original net at step 50 tick 0         (b) Steered net at step 50 tick 0         (c) Steered net at step 100 tick 0

Figure 22: The sum of activations of the box-movement channels on a $40 \times 40$ variant of the backtrack level from Figure 16 for **(a)** the original network at step 50, and the weight-steered network at **(b)** step 50 and **(c)** step 100 when the agent reaches halfway through. The original network fails to solve the level as the plan decays and cannot be extended beyond $10 - 15$ squares. Upon weight steering, the plan activations travel farther without decaying thus solving the level.

## P   LIMITATIONS

Our paper has several limitations.

We only reverse-engineer DRC and no other networks. It is possible that the inductive biases of other networks such as transformer, Conv-ResNet, or 1D-LSTM may end up learning an algorithm that is different from what we found for the DRC. Our results are also only on Sokoban and it is possible that the learned algorithm for other game-playing network looks very different from the one learned for Sokoban.

We also do not fully reverse-engineer the network. We have observed the following behaviors that cannot be explained yet with our current understanding of the learned algorithm:

- Agent sometimes executes some steps of the plan for box 1, then box 2, then back to box 1, to minimize distance. Our explanation doesn't account for how and when the network switches between boxes.
- Sometimes the heuristics inexplicably choose where to go based on seemingly irrelevant things. Slightly changing the shape or an obstacle or moving the agent's position by 1 can sometimes change which plan gets chosen, in a manner that doesn't correspond to optimal plan.

## Q   SOCIETAL IMPACT

This research into interpretability can make models more transparent, which helps in making models predictable, easier to debug and ensure they conform to specifications.

Specifically, we analyze a model organism which is planning. We hope that this will catalyze further research on identifying, evaluating and understanding what *goal* a model has. We hope that directly identifying a model's goal lets us monitor for and correct goal misgeneralization (Di Langosco et al., 2022).

Table 8: Grouped channels and their descriptions. * indicates long-term channels.

| Group | Description | Channels |
|---|---|---|
| Box up | Activates on squares from where a box would be pushed up | L0H13, L0H24*, L2H6 |
| Box down | Activates on squares from where a box would be pushed down | L0H2, L0H14*, L0H20*, L1H14*, L1H17, L1H19 |
| Box left | Activates on squares from where a box would be pushed left | L0H23*, L0H31, L1H11, L1H27, L2H20 |

Continued on next page

Table 8: Grouped channels and their descriptions. * indicates long-term channels.

| Group | Description | Channels |
|---|---|---|
| Box right | Activates on squares from where a box would be pushed right | L0H9, L0H17, L1H13, L1H15*, L2H9*, L2H15 |
| Agent up | Activates on squares from where an agent would move up | L0H18, L1H5, L1H29, L2H28, L2H29 |
| Agent down | Activates on squares from where an agent would move down | L0H10, L1H18, L2H4, L2H8 |
| Agent left | Activates on squares from where an agent would move left | L2H23, L2H27, L2H31 |
| Agent right | Activates on squares from where an agent would move right | L1H21, L1H28, L2H3, L2H5, L2H21*, L2H26 |
| Combined Plan | Channels that combine plan information from multiple directions | L0H15, L0H16, L0H28, L0H30, L1H0, L1H4, L1H8, L1H9, L1H20, L1H25, L2H0, L2H1, L2H13, L2H14, L2H17, L2H18, L0H7, L0H1, L0H21, L1H2, L1H23, L2H11, L2H22, L2H24, L2H25, L2H12, L2H16, L0H19, L2H30 |
| Entity | Highly activate on target tiles. Some also activate on agent or box tiles | L0H6, L0H26, L1H6, L1H10, L1H22, L1H31, L2H2, L2H7 |
| No label | Uninterpreted channels. These channels do not have a clear meaning but they are also not very useful | L0H0, L0H3, L0H4, L0H5, L0H8, L0H22, L0H25, L0H27, L0H29, L1H1, L1H3, L1H12, L1H16, L1H26, L1H30, L2H10, L2H19, L0H11, L0H12, L1H7, L1H24 |
| Grid-Next-Action (GNA) | Channels that activate on squares that the agent will move in the next few moves. One separate channel for each direction | L2H28 (up), L2H4 (down), L2H23 (left), L2H26 (right) |
| Pooled-Next-Action (PNA) | A channel for each action that activates highly across all squares at the last tick ($n = 2$) to predict the action | L2H29 (up), L2H8 (down), L2H27 (left), L2H3 (right) |

Table 9: Informal description of all channels

| Channel | Long-term | Description |
|---|---|---|
| L0H0 | No | some box-left-moves? |
| L0H1 | No | box-to-target-lines which light up when agent comes close to the box. |
| L0H2 | No | H/-C/-I/J/-O: +future box down moves [1sq left] |
| L0H5 | No | [1sq left] |
| L0H6 | No | H/-C: +target -box -agent . F: +agent +agent future pos. I: +agent. O: -agent future pos. J: +target -agent[same sq] |
| L0H7 | No | (0.37 corr across i,j,f,o). |
| L0H9 | No | -H/-C/-O/I/J/F: +agent +future box right moves -box. -H/J/F: +agent-near-future-down-moves [on sq] |
| L0H10 | No | H: -agent-exclusive-down-moves [1sq left,down]. Positively activates on agent-exclusive-up-moves. |
| L0H11 | No | H: CO. O: box-right moves C/I: -box future pos [1sq up (left-right noisy)] |
| L0H12 | No | H: very very faint horizontal moves (could be long-term?). I/O: future box horizontal moves (left/right). [on sq] |
| L0H13 | No | H/C/I/J/O: +future box up moves [1sq up] |

Continued on next page

Table 9: Informal description of all channels

| Channel | Long-term | Description |
|---------|-----------|-------------|
| L0H14 | Yes | H/-I/O/C/H: -future-box-down-moves. Is more future-looking than other channels in this group. Box down moves fade away as other channels also start representing them. Sometimes also activates on -agent-right-moves [on sq] |
| L0H15 | No | H/I/J/-F/-O: +box-future-moves. More specifically, +box-down-moves +box-left-moves. searchy (positive field around target). (0.42 corr across i,j,f,o). |
| L0H16 | No | H +box-right-moves (not all). High negative square when agent has to perform DRU actions. [1sq up,left] |
| L0H17 | No | H/I/J/F/O: +box-future-right moves. O: +agent [1sq up] |
| L0H18 | No | H: -agent-exclusive-up-moves |
| L0H20 | Yes | H: box down moves. Upper right corner positively activates (0.47 start -> 0.6 in a few steps -> 0.7 very later on). I: -box down moves. O: +box down moves -box horizontal moves. [1sq up] |
| L0H21 | No | -box-left-moves. +up-box-moves |
| L0H23 | Yes | H/C/I/J/O: box future left moves [1sq up,left] |
| L0H24 | Yes | H/C/I/J/O: -future box up moves. long-term because it doesn't fade away after short-term also starts firing [1sq up,left] |
| L0H26 | No | H: -agent . I/C/-O: all agent future positions. J/F: agent + target + BRwalls, [1sq up] |
| L0H28 | No | H/C/I/J/F/-O: -future box down moves (follower?) [on sq]. Also represents agent up,right,left directions (but not down). |
| L0H30 | No | H/I: future positions (0.47 corr across i,j,f,o). |
| L1H0 | No | H: -agent -agent near-future-(d/l/r)-moves + box-future-pos [on sq] |
| L1H2 | No | -box-left-moves |
| L1H4 | No | +box-left moves -box-right moves [1sq up]. |
| L1H5 | No | H: +agent-exclusive-future-up moves [2sq up, 1sq left] |
| L1H6 | No | J: player (with fainted target) |
| L1H7 | No | H: - some left box moves or right box moves (ones that end at a target)? Sometimes down moves? (unclear) |
| L1H8 | No | box-near-future-down-moves(-0.4),agent-down-moves(+0.3),box-near-future-up-moves(+0.25) [on sq] |
| L1H9 | Yes | O/I/H: future pos (mostly down?) (seems to have alternate paths as well. Ablation results in sligthly longer sols on some levels). Fence walls monotonically increase in activation across steps (tracking time). [on sq] |
| L1H10 | No | J/H/C: -box + target +agent future pos. (neglible in H) O,-I: +agent +box -agent future pos [1sq up] (very important feature – 18/20 levels changed after ablation) |
| L1H11 | No | -box-left-moves (-0.6). |
| L1H13 | No | H: box-right-moves(+0.75),agent-future-pos(+0.02) [1sq left] |
| L1H14 | Yes | H: longer-term down moves? [1sq up] |
| L1H15 | Yes | H/-O: box-right-moves-that-end-on-target (with high activations towards target). Activates highly when box is on the left side of target [on sq]. |
| L1H17 | No | H/C/I/-J/-F/O: -box-future down moves [on sq] |
| L1H18 | No | H/-O: +agent future down moves (stores alternate down moves as well?) [on sq] |
| L1H19 | No | H/-F/-J: -box-down-moves (follower?) [1sq up] |
| L1H20 | No | +near-future-all-box-moves [1sq up]. |
| L1H21 | No | H: agent-right-moves(-0.5) (includes box-right-pushes as well) |
| L1H22 | No | -target |
| L1H23 | No | -box-left-moves. |
| L1H24 | No | H: -box -agent-future-pos -agent, [1sq left] |

Table 9: Informal description of all channels

| Channel | Long-term | Description |
|---|---|---|
| L1H25 | No | all-possible-paths-leading-to-targets(-0.4),agent-near-future-pos(-0.07),walls-and-out-of-plan-sqs(+0.1),boxes(+0.6). H: +box -agent -empty -agent-future-pos \| O/-C: -agent +future sqs (probably doing search in init steps) \| I: box + agent + walls \| F: -agent future pos \| J: +box +wall -agent near-future pos [1sq up,left] |
| L1H27 | No | H: box future left moves [1sq left] |
| L1H28 | No | some-agent-exclusive-right-moves(+0.3),box-up-moves-sometimes-unclear(-0.1) |
| L1H29 | No | agent-near-future-up-moves(+0.5) (~5-10steps, includes box-up-pushes as well). I: future up moves (~almost all moves) + agent sq [1sq up] |
| L1H31 | No | H: squares above and below target (mainly above) [1sq left & maybe up] |
| L2H0 | No | -box-all-moves. |
| L2H1 | No | H/O: future-down/right-sqs [1sq up] |
| L2H2 | No | H: high activation when agent is below a box on target and similar positions. walls at the bottom also activate negatively in those positions. |
| L2H3 | No | H: +right action (PNA) + future box -down -right moves + future box +left moves |
| L2H4 | No | O: +near-future agent down moves (GNA). I: +agent/box future pos [1sq left] |
| L2H5 | No | H/C/I/J: +agent-future-right-incoming-sqs, O: agent-future-sqs [1sq up, left] |
| L2H6 | No | H: +box-up-moves (~5-10 steps). -agent-up-moves. next-target (not always) [1q left] |
| L2H7 | No | +unsolved box/target |
| L2H8 | No | down action (PNA). |
| L2H9 | Yes | H/C/I/J/O: +future box right moves [1sq up] |
| L2H11 | No | -box-left-moves(-0.15),-box-right-moves(-0.05) |
| L2H13 | No | H: +box-future-left -box-long-term-future-right(fades 5-10moves before taking right moves) moves. Sometimes blurry future box up/down moves [1sq up] |
| L2H14 | No | H: all-other-sqs(-0.4) agent-future-pos(+0.01) O: -agent-future-pos. I: +box-future-pos |
| L2H15 | No | -box-right-moves [1sq up,left] |
| L2H17 | No | H/C: target(+0.75) box-future-pos(-0.3). O: target. J: +target -agent +agent future pos. I/F: target. [1sq up] |
| L2H18 | No | box-down/left-moves(-0.2). Very noisy/unclear at the start and converges later than other box-down channels. |
| L2H19 | No | H: future agent down/right/left sqs (unclear) [1sq up] |
| L2H20 | No | H: -box future left moves [1sq left] |
| L2H21 | Yes | H: -far-future-agent-right-moves. Negatively contributes to L2H26 to remove far-future-sqs. Also represents -agent/box-down-moves. [1sq up] |
| L2H22 | No | H: box-right-moves(+0.3),box-down-moves(0.15). O future sqs |
| L2H23 | No | H: future left moves (does O store alternate left moves?) (GNA). [1sq left] |
| L2H24 | No | box-right/up-moves (long-term) |
| L2H25 | No | unclear but (8, 9) square tracks value or timesteps (it is a constant negative in the 1st half episode and steadily increases in the 2nd half)? |
| L2H26 | No | H/O: near-future right moves (GNA). [on sq] |
| L2H27 | No | left action (PNA). T0: negative agent sq with positive sqs up/left. |
| L2H28 | No | near-future up moves (GNA). O: future up moves (not perfectly though) [1sq up] |
| L2H29 | No | Max-pooled Up action channel (PNA). |
| L2H31 | No | some +agent-left-moves (includes box-left-pushes). |

Table 5: Activation offset along (row, column) in the grid for each layer and channel

|  | Layer 0 | Layer 1 | Layer 2 |
|---|---|---|---|
| Channel 0 | (1, 0) | (0, 0) | (-1, 0) |
| Channel 1 | (0, 0) | (-1, -1) | (-1, -1) |
| Channel 2 | (0, -1) | (-1, 0) | (0, 0) |
| Channel 3 | (0, 0) | (-1, 0) | (0, 0) |
| Channel 4 | (-1, -1) | (-1, -1) | (0, -1) |
| Channel 5 | (0, -1) | (-2, -1) | (-1, 0) |
| Channel 6 | (0, 0) | (-1, -1) | (-1, -1) |
| Channel 7 | (-1, 0) | (-1, 0) | (0, 0) |
| Channel 8 | (0, -1) | (0, 0) | (-1, 0) |
| Channel 9 | (0, 0) | (0, 0) | (-1, 0) |
| Channel 10 | (-1, -1) | (-1, 0) | (-1, 0) |
| Channel 11 | (-1, 0) | (0, -1) | (0, -1) |
| Channel 12 | (0, -1) | (0, -1) | (0, -1) |
| Channel 13 | (-1, 0) | (-1, 0) | (-1, 0) |
| Channel 14 | (0, 0) | (0, -1) | (-1, -1) |
| Channel 15 | (0, 0) | (0, 0) | (-1, -1) |
| Channel 16 | (-1, -1) | (0, 0) | (-1, -1) |
| Channel 17 | (-1, 0) | (0, 0) | (-1, 0) |
| Channel 18 | (-1, 0) | (0, 0) | (-1, 0) |
| Channel 19 | (-1, -1) | (-1, 0) | (-1, -1) |
| Channel 20 | (-1, 0) | (0, -1) | (0, -1) |
| Channel 21 | (-1, 0) | (-1, 0) | (0, 0) |
| Channel 22 | (0, 0) | (0, 0) | (-1, 0) |
| Channel 23 | (-1, -1) | (-1, 0) | (0, -1) |
| Channel 24 | (-1, -1) | (0, -1) | (-1, 0) |
| Channel 25 | (-1, 0) | (-1, -1) | (-1, -1) |
| Channel 26 | (-1, 0) | (0, -1) | (0, 0) |
| Channel 27 | (-1, -1) | (-1, -1) | (0, 0) |
| Channel 28 | (0, 0) | (0, 0) | (-1, 0) |
| Channel 29 | (0, 0) | (-1, 0) | (0, -1) |
| Channel 30 | (-1, 0) | (0, 0) | (-1, -1) |
| Channel 31 | (-1, -1) | (0, -1) | (0, -1) |

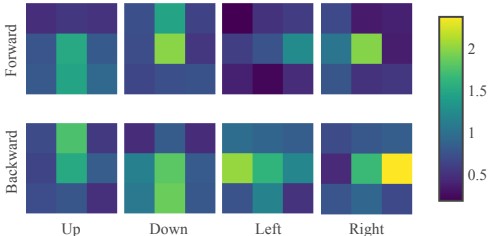

(a) Forward and backward plan extension kernels averaged over agent-movement channels. Agent-movement channels also extend the agent moves forward and backward similar to the box-plan extension.

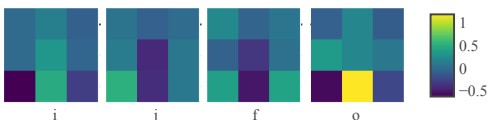

(b) The kernels that map L1H17 (box-down) to L1H18 (agent-down) by shifting the activation one square up. L1H17 activates negatively, therefore the $j$ and $f$ kernels are negative since they use the sigmoid activation function. The $i$ and $o$ kernels are positive which results in negatively activating $i$ and $o$-gates, which after multiplication results in L1H18 activating positively. The opposite signed weights on the lower-corner squares of the kernel help in picking a single path out of multiple parallel paths.

Figure 23: Plan extension and box path to agent path kernels.

Table 6: Correlation of linear regression model's predictions with the original activations for each channel.

|  | Layer 0 | Layer 1 | Layer 2 |
|---|---|---|---|
| Channel 0 | 33.15 | 79.48 | 70.03 |
| Channel 1 | 50.76 | 48.77 | 38.37 |
| Channel 2 | 73.15 | 28.90 | 39.17 |
| Channel 3 | 31.73 | 68.30 | 55.72 |
| Channel 4 | 45.06 | 50.10 | 45.64 |
| Channel 5 | 63.91 | 42.95 | 55.27 |
| Channel 6 | 96.57 | 87.47 | 53.90 |
| Channel 7 | 51.98 | 36.88 | 95.63 |
| Channel 8 | 46.64 | 41.58 | 55.04 |
| Channel 9 | 70.52 | 37.44 | 71.47 |
| Channel 10 | 37.68 | 99.01 | 53.91 |
| Channel 11 | 52.09 | 61.55 | 42.26 |
| Channel 12 | 41.54 | 43.86 | 27.19 |
| Channel 13 | 79.54 | 73.35 | 54.40 |
| Channel 14 | 72.17 | 48.12 | 56.54 |
| Channel 15 | 44.09 | 65.72 | 36.37 |
| Channel 16 | 63.49 | 26.56 | 38.24 |
| Channel 17 | 76.70 | 73.94 | 94.78 |
| Channel 18 | 61.51 | 66.11 | 34.18 |
| Channel 19 | 46.05 | 44.01 | 33.48 |
| Channel 20 | 65.00 | 58.94 | 64.92 |
| Channel 21 | 22.05 | 57.36 | 60.21 |
| Channel 22 | 26.51 | 63.73 | 24.32 |
| Channel 23 | 74.39 | 31.32 | 44.64 |
| Channel 24 | 83.64 | 58.56 | 59.94 |
| Channel 25 | 17.10 | 82.43 | 28.29 |
| Channel 26 | 75.48 | 44.26 | 45.17 |
| Channel 27 | 9.24 | 85.84 | 49.92 |
| Channel 28 | 46.87 | 42.65 | 15.38 |
| Channel 29 | 28.60 | 64.77 | 54.68 |
| Channel 30 | 47.70 | 35.00 | 40.15 |
| Channel 31 | 53.12 | 56.81 | 59.63 |

Table 7: Correlation of linear regression model's predictions with the original activations averaged over channels for each group. Includes correlation using only base features for comparison. The (all dir) group is the average of the four directions. NGA and PNA are included in the Agent groups.

| Group | Correlation | Base correlation |
|---|---|---|
| Box up | 72.36 | 21.01 |
| Box down | 62.73 | 13.93 |
| Box left | 67.96 | 21.10 |
| Box right | 65.69 | 27.40 |
| Box (all dir) | 66.37 | 20.83 |
| Agent up | 47.86 | 12.69 |
| Agent down | 51.12 | 15.85 |
| Agent left | 51.40 | 7.85 |
| Agent right | 52.73 | 14.92 |
| Agent (all dir) | 50.80 | 13.33 |
| Combined path | 48.00 | 23.35 |
| Entity | 76.73 | 70.66 |
| No label | 40.25 | 15.53 |

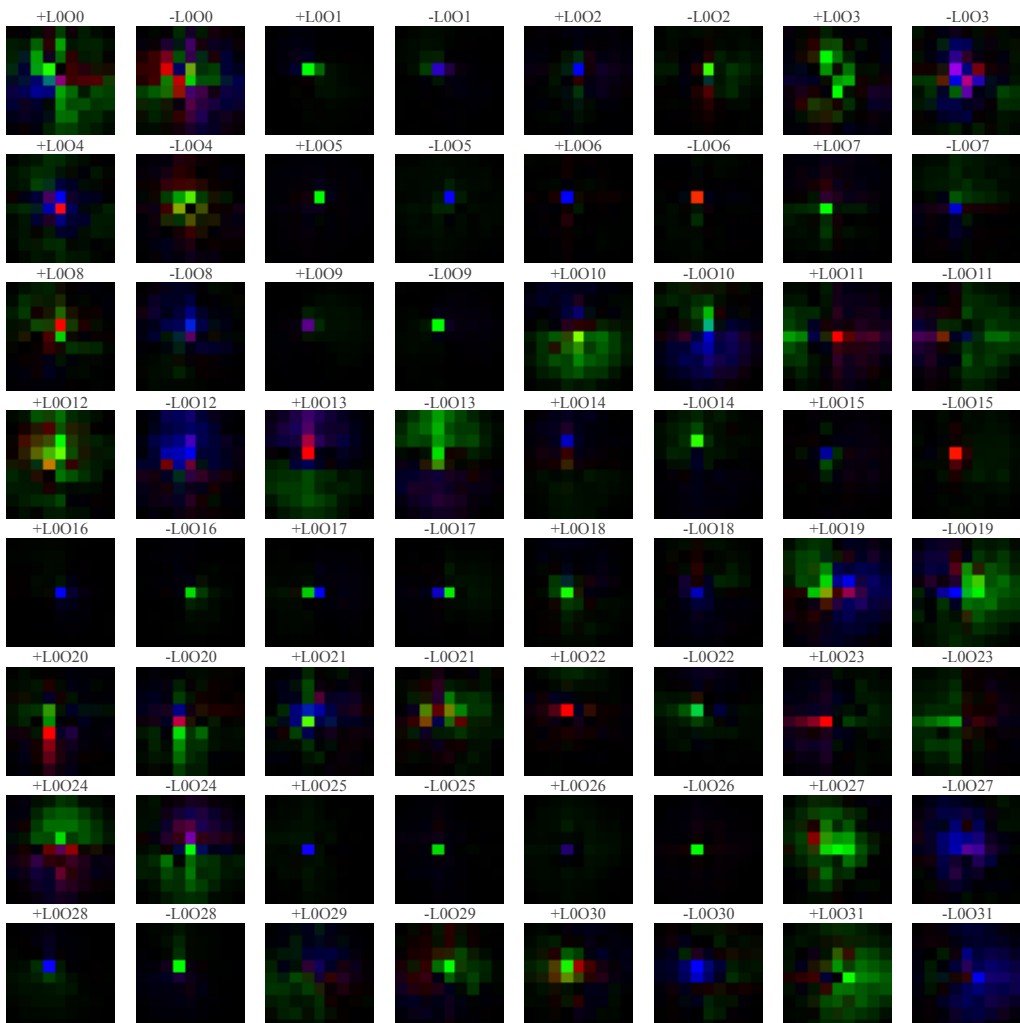

Figure 24: $9 \times 9$ combined convolutional filters $W_{oe}^0$ that map the RGB observation image to the $O$ gate in layer 0. The positive and negative components of each channel filters are separated visualized by computing $max(0, W_{oe}^0)$ and $max(0, -W_{oe}^0)$ respectively. The green, red, and brown colors in the filters detect the agent, target, and box squares respectively. The blue component is high only in empty tiles, so the blue color can detect empty tiles. We find that many filters are responsible for detecting the agent and the target like L0O5 and L0O6. A use case of such agent and box detecting filters in the encoder is shown in Figure 8. Many filters detect whether the agent or the target are some squares away in a particular direction like L0O20 and L0O23. Filters for other layers and gates can be visualized using our codebase.

Table 10: Solve rate (%) of different models without and with 6 thinking steps on held out sets of varying difficulty.

| Model | No Thinking | | | Thinking | | |
|---|---|---|---|---|---|---|
| | Hard | Med | Unfil | Hard | Med | Unfil |
| DRC(3, 3) | 42.8 | 76.6 | 99.3 | 49.7 | 81.3 | 99.7 |
| DRC(1, 1) | 7.8 | 28.1 | 89.4 | 9.8 | 33.9 | 92.6 |
| ResNet | 26.2 | 59.4 | 97.9 | - | - | - |

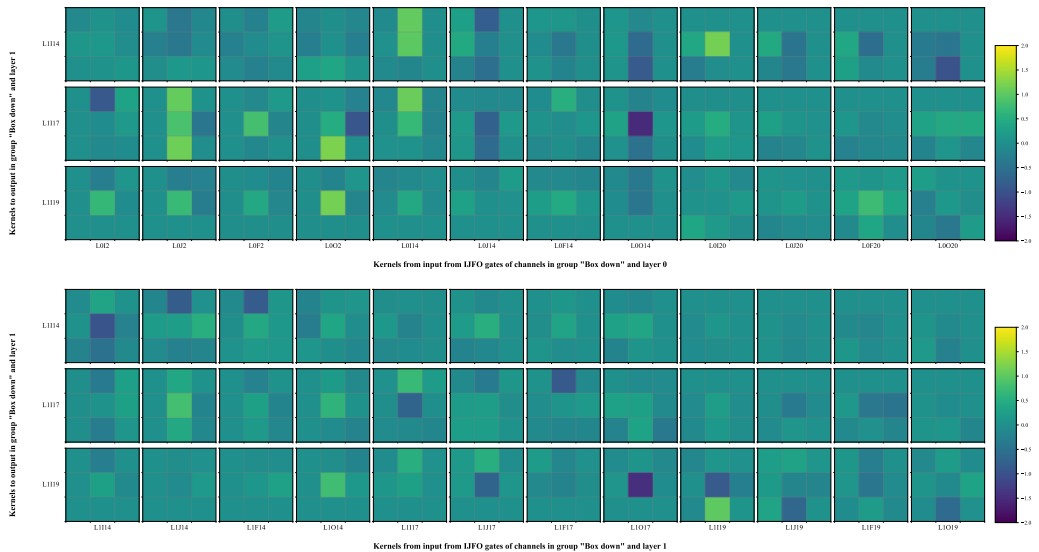

Figure 25: Each of the Box-down to box-down plan-extension kernels, centered by their channels' relative offsets (see Table 5). The first 3 rows are kernels from IJFO of layer 0 to H of layer 1, and the next 3 from IJFO of layer 1 to the H in its next step. In many cases we see the idealized weight pattern from Figure 7a, but in most we do not. The color scale goes from −2.0 to 2.0.

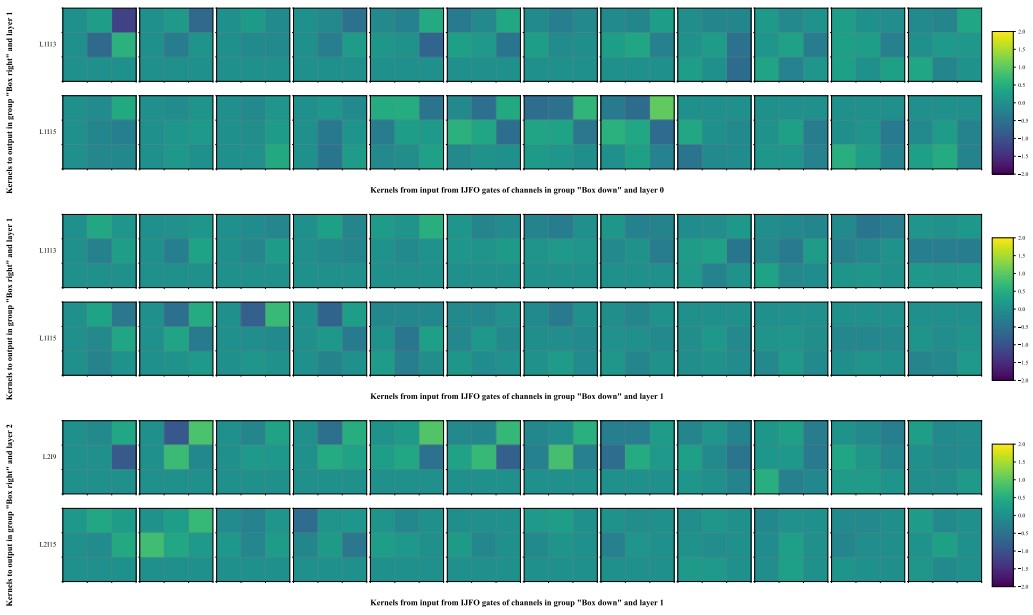

Figure 26: Each of the Box-down to box-down plan-extension kernels, centered by their channels' relative offsets (see Table 5). The first two rows are kernels from IJFO of layer 0 to H of layer 1, the middle two from IJFO of layer 1 to the H in its next step, and the last two from IJFO of layer 1 to H of layer 2. In many cases we see the idealized weight pattern from Figure 7a, but in most we do not. The color scale goes from −2.0 to 2.0.

