# OpenReview forum: "Path Channels and Plan Extension Kernels: a Mechanistic Description of Planning in a Sokoban RNN"
_ICLR.cc/2026/Conference — ICLR 2026 Poster_

### Official Review · Reviewer_a388 · 2025-11-01

**Soundness:** 3
**Presentation:** 3
**Contribution:** 3
**Rating:** 6
**Confidence:** 2

**Summary:**

Sokoban is a 2d grid based game where the aim is to push blocks onto targets.  However, not all moves are reversible, meaning that solutions that work need a understand the longer time consequences of their actions to ensure they don't get stuck.  Deep Repeated ConvLSTM (DRC) is a Recurrent Neural Networks (RNN) model that can learn to play Sokoban. (Bush et al. 2004) demonstrated that that DRC is performing planning by using linear probes on the activations to predict the long term effects of its actions.  They also showed that the plans proceed both forward from the agent position and backward from the targets, and that it was possible via interventions to affect the plan that the agent took.

The current paper goes into considerably more depth about how the network is actually representing future plans, including assigning interpretable meanings to many of the different channels of the DRC network as opposed to only using linear probes.  This includes finding that each channel encodes a propensity to move in a specific direction.  They also separate out box movement and agent movement channels. Note that not all channels are interpretable, and the full meaning of the redundant layers of each type is not fully discernible either.  Furthermore, they show how the network can deal with overlapping channels, where the same grid square is used at different time points by highlighting the activations at different times.

They also dig into the specific ways the convolutional kernels are used to build up the plan over inference time (not agent time), including being blocked by walls and what they term Turn Plan Extensions.

They use this knowledge in a variety of ways, like scaling the magnitude of certain weights to scale to larger maps, and like Bush et al, intervening to make the agent take sub-optimal plans.

Furthermore they demonstrate the winner takes all mechanism where short term plans suppress each other.

**Strengths:**

## Originality

The work is incremental on top of Bush et al, 2004, but it does go deeper into explaining how an RNN can solve the planning problem.  This level of detailed analysis of a planning RNN is to my knowledge an original contribution

## Quality

It's not a fully generative description of the RNN - the knowledge they find is not sufficient to build one from first principles, so it is only a partial understanding.  This in turn leads to the paper begin a bit partial, like the knowledge they find.  However, the hypothesis they generate are generally well tested empirically and illustrated well in the paper.

## Clarity
The paper is generally well written, and it would be straightforward to replicate their results and experiments.

## Significance
Planning is a vital step in understanding and controlling the world.  The current paper, while it is not a fully integrated solution may well lead to further insights into how planning is both learnt and performed in other systems like LLMs as well, which would be a very significant potential future impact.

**Weaknesses:**

As said before, it's not a full explanation of how the DRC networks works, it still provides various valuable insights into how different layers work.  Obviously being able to construct a simplified planning network given the knowledge would be a very powerful contribution, but that is outside of scope of this paper, and this paper is a step towards that goal.

**Questions:**

Did you examine any intermediate checkpoints to see when the different behaviors emerged?

---

> ### Author Response · Authors · 2025-12-04
>
> We thank the reviewer for the thoughtful evaluation and for recognizing our contribution as "an original contribution" that "may well lead to further insights into how planning is both learnt and performed in other systems like LLMs."
>
> ### Q: Did you examine intermediate checkpoints to see when the different behaviors emerged?
>
> We have now analyzed intermediate checkpoints. We’ve added a new Appendix M titled “Emergence of planning structure during training.” We perform an experiment where we track the emergence of the WTA kernels through the 56 checkpoints stored during training of the main network. We show that the WTA kernels emerge around 19M environmental steps when the path channels begin to reinforce themselves while inhibiting channels in other directions. This effect then strengthens throughout further training steps.
>
> ### Clarification: Contribution Beyond Bush et al. (2025)
>
> We appreciate the opportunity to clarify how our work extends beyond prior work. While the reviewer characterizes our contribution as incremental, we believe the advances are more substantial:
>
> **From representation to mechanism.** Bush et al. discovered that plans can be decoded from activations using learned linear probes. We discover how plans are computed—the specific channels, kernels, and operations implementing planning. This is analogous to the difference between showing a brain region correlates with a behavior versus explaining the circuit mechanism.
>
> | Aspect | Bush et al. (2025) | This work |
> | -- | -- | -- |
> | Plan access | Linear probes (learned) | Direct channel reading |
> | Causal intervention | 82.5% (box), 20.7% (agent) | 99.7% (PNA channels) |
> | Bidirectional search | Qualitative observation | Mechanistic: kernels in Fig. 7 |
> | Path selection | Not addressed | WTA mechanism (Fig. 10) |
> | Backtracking | Not addressed | Negative activation propagation (Sec. 3.3) |
> | Weight-level analysis | None | Intervention and ablation experiments on extension and WTA kernels |
>
> **Concrete mechanistic discoveries not present in prior work:**
> 1. _Plan extension kernels (Section 3.2, Fig. 7):_ We identify the specific 3×3 convolutional kernels that extend paths forward and backward, showing they encode a learned transition model.
> 2. _Winner-takes-all mechanism (Section 3.3, Fig. 10):_ We show how competing paths are resolved through inhibitory connections between short-term path channels, with causal verification via kernel ablation (Fig. 9).
> 3. _Backtracking via negative propagation (Section 3.3, Fig. 17):_ We demonstrate that walls generate negative activations that propagate backward along paths, pruning invalid plans—verified with 85.1% intervention success rate on long-term channels.
> 4. _Weight steering for generalization (Section L):_ Our mechanistic understanding enabled a novel intervention—scaling plan extension kernels by 1.4× allows solving 40×40 levels despite training only on 10×10.
>
> ### On Partial vs. Complete Understanding
> The reviewer notes this is "not a fully generative description." We agree and have been transparent about limitations (Section N). However, we emphasize:
> * 78% of channels are interpretable (75/96), with the remaining 21 "no-label" channels showing low importance in ablations
> * The mechanisms form a coherent algorithm: initialization → extension → WTA → backtracking → action selection
> * This advances the Pareto frontier between network complexity and explanation detail (Section 4), providing one of the most comprehensive description of a neural network of this complexity.
>
> ---
>
> We hope this clarifies the significance of our mechanistic contributions and answers questions about the emergence of the channels and the kernels. We are happy to address any further questions.

---

### Official Review · Reviewer_UMsf · 2025-11-01

**Soundness:** 3
**Presentation:** 2
**Contribution:** 3
**Rating:** 4
**Confidence:** 3

**Summary:**

This paper analyzes a Sokoban-trained DRC(3,3) agent and shows that it performs internal planning through path channels and plan extension kernels. Using ablation and causal tests, the authors demonstrate that these components are responsible for planning behavior, providing a clear mechanistic view of how model-free networks can develop search-like computation.

**Strengths:**

This paper offers a detailed mechanistic analysis of the DRC(3,3) reinforcement learning agent trained on Sokoban. Its originality lies in the depth of interpretability achieved—the authors go beyond linear probes to reveal path channels and plan extension kernels that implement a neural form of bidirectional planning.

The study combines qualitative visualization, quantitative ablation, and causal intervention experiments in a careful way. The causal and ablation results convincingly show that these channels are genuinely responsible for planning.

Overall, it provides strong evidence that a model-free ConvLSTM can internally develop search-like planning dynamics, offering a valuable model system for studying emergent planning and mesa-optimization.

**Weaknesses:**

Despite its depth, the paper can be very difficult to read.  Readers unfamiliar with DRC(3,3) may struggle to follow the layer/tick conventions and channel indexing. A schematic overview early in the paper would improve clarity.

Another issue is limited generality: all findings are restricted to Sokoban. It remains unclear whether the same mechanisms emerge in other planning-heavy environments or different architectures. Some comparative evidence (e.g., other environments) would help validate the universality of these “path channels.”

The causal analysis is strong but statistical validation could be deeper: no confidence intervals or significance testing accompany success-rate reductions. Moreover, it is unclear whether the WTA mechanism and negative activation propagation are emergent or enforced by architectural biases. Explicitly disentangling inductive structure from emergent behavior would strengthen the argument.


Some other isseus:
1. Many references in the paper are broken. Line 834, Line 1050, Line 1215
2. I don't understand the "?" in Table 8.
3. Figure 15 is broken.
4. Line 041: agent's goal [rectange], should be "agent's [rectange] goal"?
5. L131: locationns.
6. Line 20: What do you mean by "parst"?

**Questions:**

When multiple boxes are present, how exactly does the network choose which one to move first? The WTA mechanism seems to operate locally across directions, but does it also act across different entities (boxes)?

Could similar path channel dynamics be found in agents trained on non-grid planning tasks (e.g., MiniGrid, navigation, or graph problems)?

How stable are the discovered channels across random seeds or retraining runs—are their spatial roles (e.g., “box-right”) consistent?

---

> ### Author Response · Authors · 2025-12-04
>
> We thank the reviewer for the careful reading and for recognizing our work as "a valuable model system for studying emergent planning and mesa-optimization." We address each concern below, with references to the revised draft where applicable.
>
> ### Weaknesses
> > Despite its depth, the paper can be very difficult to read. Readers unfamiliar with DRC(3,3) may struggle to follow the layer/tick conventions and channel indexing. A schematic overview early in the paper would improve clarity.
>
> Good idea! We've now added such a figure (Section 3.1, Figure 3) with a simplified architecture diagram with an explanation of the layer/tick setup.
>
> > Another issue is limited generality: all findings are restricted to Sokoban. It remains unclear whether the same mechanisms emerge in other planning-heavy environments or different architectures. Some comparative evidence (e.g., other environments) would help validate the universality of these “path channels.”
>
> The original Guez et. al paper (and subsequent literature, including Bush et. al and Taufeeque et. al) only demonstrate the learned planning behavior in the Sokoban setting, which we focus on. We follow the mechanistic interpretability motivation of the literature: trying to deeply understand how one network works as a starting point for understanding others. This mirrors successful mechanistic interpretability work (e.g., curve circuits in InceptionV1, IOI in GPT-2). We agree that investigating the universality of path channels would be an interesting direction for follow-up work.
>
> > The causal analysis is strong but statistical validation could be deeper: no confidence intervals or significance testing accompany success-rate reductions.
>
> We have now added 95% confidence intervals to the point estimates in Table 3 and Section 2.3, via bootstrap. All our results, tables, and figures now have 95% confidence intervals.
>
> > Moreover, it is unclear whether the WTA mechanism and negative activation propagation are emergent or enforced by architectural biases. Explicitly disentangling inductive structure from emergent behavior would strengthen the argument.
>
> This is a good question! The DRC (3, 3) architecture is designed to have recurrent structure that allows for activations to be continually developed over time, and think that interesting follow up work could see if other architectures (particularly transformers) also have this structure.
>
> > Some other issues...
>
> Thanks for the feedback, we have fixed the various typos.
>
> ### Questions:
> > When multiple boxes are present, how exactly does the network choose which one to move first? The WTA mechanism seems to operate locally across directions, but does it also act across different entities (boxes)?
>
> Good question! As briefly mentioned in the Discussion section "Path channel activations as a value function", the choice of which direction to go in first appears to be implicit in the network. The ticks and convolutional structure are only able to propagate plan segments at 3 squares per timestep, bidirectionally extended from the agent and boxes. The agent follows the first path which reaches it and stabilizes.
>
> > Could similar path channel dynamics be found in agents trained on non-grid planning tasks (e.g., MiniGrid, navigation, or graph problems)?
>
> Possibly! We think that this would be a promising direction for follow-up work. The path channels could be recurring in planning-based settings, or only need to emerge in more complicated settings with complex dependencies between steps. For example, a simple maze task may not need a long vs. short term path channel distinction, because the agent doesn't need to return to previous positions and take different actions.
>
> >How stable are the discovered channels across random seeds or retraining runs—are their spatial roles (e.g., “box-right”) consistent?
>
> Since the box move channels were the most causally impactful, we've now added an analysis box move channels for additional seeds. We find qualitatively similar results, with up/down/left/right spatial roles, as well as long- and short-term path channels.

---

### Official Review · Reviewer_oYwH · 2025-11-01

**Soundness:** 4
**Presentation:** 4
**Contribution:** 3
**Rating:** 6
**Confidence:** 3

**Summary:**

This paper provides an important case study that
manually inspects the weights and the activations of a successful RNN-based Sokoban solver
that was trained by reinforcement learning.
The authors showed that, through manual inspection of activations and weights,

-   some layers model possible paths of the boxes (path channels)
-   some convolutional kernels will extend the paths from the init and the goal, as if running a bidirectional search (plan extension kernel)
-   resulting paths are rejected by winner-takes-all mechanism

These claims are checked qualitatively and quantitatively using various tools, e.g., by causal analysis, or by measuring the overlap between
the actual movable locations of the boxes and the predicted locations in the path channels.

This is a unique paper in the modern era; I have never read a paper like this,
to be honest, although I know the early NN literature has full of papers like this that
perform a similar manual probing on a CNN for handwritten digit recognition.

This paper may hype. However I don't know if Neurips is appropriate for this kind of paper,
because, to me, conference papers are usually about technical contributions.
I really don't know how to review or score this paper unlike other papers.
I admit I might be narrow-minded, and I prefer to defer the decision to SPCs/ACs.
I apologize that my review is unusually short despite being from the planning background.

**Strengths:**

I really do not have much to say.

It may have some implications on the RL-trained the language models,
especially those that reuse the same layer several times (e.g. diffusion language models, Universal Transformer, Sparse Universal Transformer)
and use think tags.

**Weaknesses:**

The analysis is performed on a particular set of weights of a particular RNN architecture using a particular dataset, random seed, etc.,
leaving it unclear whether the findings generalize to wider applications,
or even to a different set of weights from a different random seed.
(the authors acknowledged this in the appendix)

The paper does not have a technical contribution.
Rather, the authors simply manually found the existence of path channels,
which renders the use of linear probes / logistic regression probes unnecessary.

I still think it is a bit of a stretch to call the bidirectional path extension / retraction behavior as "running a search algorithm".
It is clearly not running a systematic tree search;
It may not backtrack when 3 directions all failed,
and it may incorrectly backtrack early before trying all 3 directions.

**Questions:**

A natural next step of this paper might be this:
What would happen if you manually bake in a similar kernel into the RNN during the weight initialization
and optionally fixing those weights?

---

> ### Author Response · Authors · 2025-12-03
>
> Thank you for your review!
>
> > However I don't know if Neurips is appropriate for this kind of paper, because, to me, conference papers are usually about technical contributions. I really don't know how to review or score this paper unlike other papers. I admit I might be narrow-minded, and I prefer to defer the decision to SPCs/ACs.
>
> We appreciate the reviewer's candor about the difficulty of evaluating this type of work. A similar paper (Thomas Bush, Stephen Chung, Usman Anwar, Adrià Garriga-Alonso, David Krueger  _“Interpreting Emergent Planning in Model-Free Reinforcement Learning”_ (ICLR 2025)) covering the Sokoban learned planner was accepted as an oral presentation at last year’s ICLR, so we believe that ICLR is an appropriate venue for our work. That paper similarly analyzes a DRC(3,3) agent on Sokoban, and improves our understanding of how it functions.
>
> For the International Conference on Learning Representations in particular, we believe that our detailed analysis of the internal state _representation_ used by an RNN-based agent with planning behavior in a complex puzzle setting is a worthwhile and interesting contribution.
>
> > The analysis is performed on a particular set of weights of a particular RNN architecture using a particular dataset, random seed, etc., leaving it unclear whether the findings generalize to wider applications, or even to a different set of weights from a different random seed. (the authors acknowledged this in the appendix)
>
> We’ve now added an analysis of 4 additional random seeds using an automated classification procedure, finding qualitatively similar results.
>
> > The paper does not have a technical contribution. Rather, the authors simply manually found the existence of path channels, which renders the use of linear probes / logistic regression probes unnecessary.
>
> While our current version uses a simple method to automatically discover path channels on additional seeds, we see our core technical contribution as unpacking how the DRC(3, 3) agent can mechanistically use path channel activations to construct plans, and the value of taking particular actions.
>
> > I still think it is a bit of a stretch to call the bidirectional path extension / retraction behavior as "running a search algorithm". It is clearly not running a systematic tree search; It may not backtrack when 3 directions all failed, and it may incorrectly backtrack early before trying all 3 directions.
>
> While the agent likely does not follow a specific planning algorithm exactly, we follow the literature (Guez et. al, Bush et. al, Taufeeque et. al) in summarizing the agent as having planning and search-like behavior. In particular, earlier work in this vein emphasized the fact that the DRC(3, 3) agent is able to benefit from additional compute in the shape of forced (Guez et. al) or emergent (Taufeeque et. al) NO-OP steps, and that its planning procedure appears to be similar to bidirectional search (Bush et. al) in particular.

---

### Official Review · Reviewer_DRUr · 2025-11-03

**Soundness:** 3
**Presentation:** 3
**Contribution:** 4
**Rating:** 8
**Confidence:** 4

**Summary:**

The paper reverse-engineers a Sokoban DRC(3,3) ConvLSTM agent trained with model-free RL and shows that its hidden state contains path channels, which directly encode future moves for boxes/agent by direction. Planning emerges via plan-extension kernels that (i) initialize short path fragments near boxes/targets from encoder features, (ii) extend them forward from boxes and backward from targets, (iii) stop at obstacles via negative activations, and (iv) use a winner-takes-all inhibition to select between competing short-term moves. Quantitatively: ablating 59 labeled path channels drops solve rate by 57.6% vs 10.5% for 37 non-path channels; random 37-path-channel ablation: 41.3%. Path channels are predictive of future moves (AUC curves separate short- vs long-term). Causal interventions on PNA/GNA channels achieve ~99% action flips; movement channels ~88% (agent-movement lower due to condition mismatch). Weight-level analysis also explains backtracking (negative activation propagation) and shows “weight steering” (scaling extension kernels) stabilizes longer paths on larger boards. The authors argue this constitutes a concrete, mechanistic description of bidirectional planning in a model-free agent and discuss links to mesa-optimization.

**Strengths:**

1. Plans live in identifiable channels; no probes needed to find them.
2. Concrete kernels for initialize/extend/stop/compete; elegant WTA tying to action selection.
3. Targeted edits flip actions at ~99% via GNA/PNA; large drops from path-channel ablations.
4. Long/short-term split with j-gate-mediated transfer explains overlapping plans.
5. The “weight steering” generalization and a clean recipe others can reuse.

**Weaknesses:**

1. Risk of confirmation bias; limited inter-rater reliability reporting.
2. Define solve-rate protocol, seeds, and statistical tests more precisely; report CIs on all numbers.
3. One architecture (DRC(3,3)), one domain (Sokoban).
4. No check on other DRC sizes, Atari-DRC, or different training runs.
5. Interesting, but current evidence is circumstantial; could be toned down or supported with extra tests (e.g., explicit internal objective readouts).
6. Success metric is “any alternate action”; consider targeted action flips and off-policy rollouts to quantify downstream task impact.

**Questions:**

1. How reproducible are channel group assignments? Provide criteria + agreement stats; release per-channel labels.
2. Does the same path-kernel story hold for different DRC depths/widths, training seeds, and other grid sizes without weight steering?
3. If you zero WTA connections, what is the task-level effect on solve rate and branching? Similarly, for turning vs linear extension kernels.
4. Report intervention success vs magnitude; do small signed nudges suffice to steer plans reliably?
5. Can you predict the critic head from path-channel activations (counts/energies) across states? Provide R^2/ablation on the value head.
6. How often do negative “stop” signals incorrectly prune valid paths? Any systematic failure modes (e.g., tunnels, multi-box couplings)?
7. Please release weights, code for kernel visualizations, and the exact Boxoban splits used.

---

> ### Author Response · Authors · 2025-12-04
>
> We thank the reviewer for their thoughtful review and detailed engagement with our work!
>
> ### Questions
> 1. Appendix N now uses a simple automated method for assigning channel labels using AUC values. This method We find that the automatic discovery method correctly labels 3/4 of the GNA channels, all the PNA channels, and discovers the box or agent channels with a F1-score of 73.7%.
> 2. The same basic results (long- and short-term path channels, extension kernels, and a winner takes all mechanism) holds for 4 other training seeds. Without weight steering, the model generalizes to larger levels, but often struggles to complete levels which are much bigger than the ones it was trained on. Figure 3a of Guez et. al shows that the DRC(3, 3) architecture is much more effective at solving levels than other variants, and that suggesting that a DRC(1, 1) architecture may not be successful at solving Sokoban levels at all.
> 3. We have not analyzed the overall solve rate of ablating different kernels, however we do have various single-level demonstrations where ablating the relevant kernels breaks the agent’s ability to solve the level.
> 4. We have not analyzed the intervention success rate vs. magnitude. Since we the DRC uses tanh nonlinearity larger magnitude changes are unlikely to change the results, and we have not investigated smaller ones.
> 5. Taufeeque et. al (2024) trains a probe to predict the critic head values, and is able to obtain a 97.7-99.7% R^2 value on this output using all channel activations.
> 6. Negative stop signals could potentially incorrectly prune paths, but we don’t have a measurement of how often this occurs. Pruned paths coils also get potentially added back even after cessation – since the plan extension kernels add activation going forward from the network, as long as a valid plan segments finds an alternative route back to the agent’s path segments, it could restabilize.
> 7. We will upload this in the supplementary material.

---

### Author Response · Authors · 2025-12-04

To the AC: Thank you for stepping up as AC for this paper in the wake of the ICLR deanonymization situation. Our primary improvement was to create and use an automated channel label assignment method to show that our overall findings of long- and short-term path channels, plan extension kernels, and a winner takes all mechanism generalizes to four other training seeds. This shows that, for the DRC(3, 3) agent typically studied in the learned planning literature, our proposed mechanisms hold across different training runs.

We have also made various other clarifications and improvements, such as adding 95% confidence intervals, illustrative figures, and the like; as shown below.

---

### Meta-Review · Area_Chair_3hAH · 2026-01-05

**Summary:**

This paper interprets/explains the neural network weights of a deep RL agent (ConvLSTM) trained to play Sokoban. This paper is quite unique in terms of its contributions, aiming to provide an understanding of solutions learned by a non-linear function approximator. It goes on to show many phenomena, including "path channels" that encode future moves, and how convolutional kernels can be seen as capturing part of a learned transition model. I personally find this type of work quite important for going beyond the traditional type of paper we see at places like ICLR. I find such a new perspective very welcome, and given how the reviewers were fairly positive about this paper, I am recommending it to be accepted.

**Reviewer Concerns:**

- _The specificity of the claims. Are the observed phenomena reproducible across different architectures?_

	The authors have reproduced them across seeds and have argued how the studied architecture makes sense. This is, in my own opinion, maybe the main limitation of the work.

- _Only a single environment (Sokoban) was investigated._

	I personally don't agree with this criticism, as one can always ask for more, and the findings are interesting in the setting they are presented.

- More clarity over statistical claims.

	As usual, the authors claim to have added such information to the new version of the paper.

**Reviewer Scores:**

- Reviewer DRUr: It is hard to imagine they would further raise their score (it is already an 8).
- Reviewer oYwH: They were unsure about the fit of the paper, so as much as they like the work, I suspect they wouldn't be comfortable raising their score (6->8).
- Reviewer UMsf: They had concerns about the presentation and the generality. I suspect they would have kept their score (4) since not much was done in terms of a new version.
- Reviewer a388: I also suspect they would have kept their score (6).

---

### Decision · Program_Chairs · 2026-01-26

Accept (Poster)